

# A plane defect in the 3d O(N) model

**Abijith Krishnan and Max A. Metlitski**[⋆]

Department of Physics, Massachusetts Institute of Technology, Cambridge, MA 02139, USA

⋆ mmetlits@mit.edu

## Abstract

It was recently found that the classical 3d O($N$) model in the semi-infinite geometry can exhibit an "extraordinary-log" boundary universality class, where the spin-spin correlation function on the boundary falls off as $\langle \vec{S}(x) \cdot \vec{S}(0) \rangle \sim \frac{1}{(\log x)^q}$. This universality class exists for a range $2 \leq N < N_c$ and Monte-Carlo simulations and conformal bootstrap indicate $N_c > 3$. In this work, we extend this result to the 3d O($N$) model in an infinite geometry with a plane defect. We use renormalization group (RG) to show that in this case the extraordinary-log universality class is present for any finite $N \geq 2$. We additionally show, in agreement with our RG analysis, that the line of defect fixed points which is present at $N = \infty$ is lifted to the ordinary, special (no defect) and extraordinary-log universality classes by $1/N$ corrections. We study the "central charge" $a$ for the O($N$) model in the boundary and interface geometries and provide a non-trivial detailed check of an $a$-theorem by Jensen and O'Bannon. Finally, we revisit the problem of the O($N$) model in the semi-infinite geometry. We find evidence that at $N = N_c$ the extraordinary and special fixed points annihilate and only the ordinary fixed point is left for $N > N_c$.

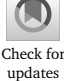

# 1 Introduction

Recent interest in the physics of boundaries and defects has been driven in part by the field of topological phases, in which such defects often expose protected modes. While the implications of bulk topological physics for defect modes are well-understood for a gapped bulk, less is known about behavior of defects and boundaries in the presence of a critical bulk. Even in the classical 3d O($N$) model, the phase diagram in the presence of a boundary or defect is not fully settled. [1]

Let us briefly review recent developments in the boundary physics of the 3d O($N$) model.

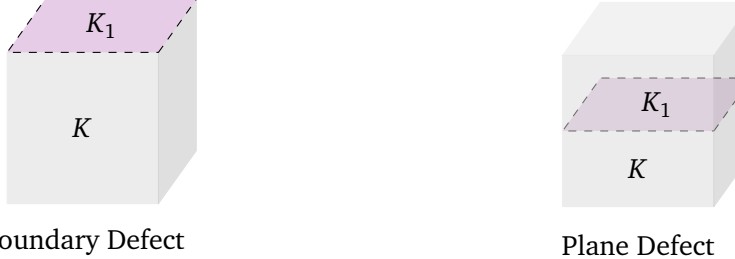

Boundary Defect          Plane Defect

Figure 1: The geometry of the boundary defect (left) and plane defect (right) for the 3d O($N$) model.

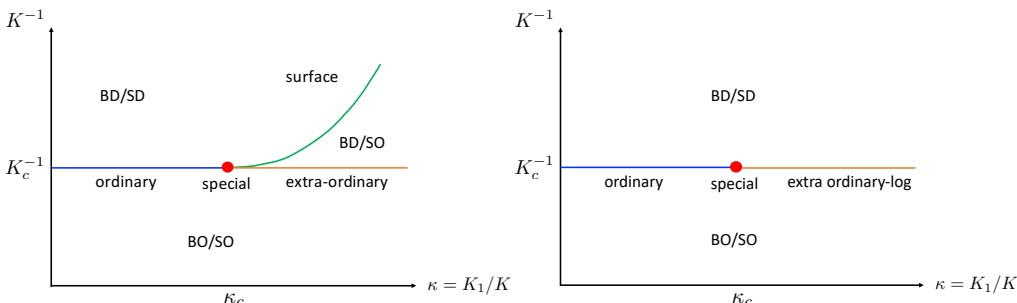

Figure 2: Left: phase diagram of the 3d O($N$) model with $N = 1, 2$ in the presence of a boundary/plane defect. BO stands for bulk ordered, SO - surface ordered, BD - bulk disordered, SD - surface disordered. For $N = 2$ the BD/SO phase only has quasi-long-range order.
Right: phase diagram of the 3d O($N$) model with a boundary for $2 < N < N_c$ or for a plane defect with $N > 2$.

Consider a system of classical $N$-component spins $\vec{S}_i$ of length one on sites of a semi-infinite 3d cubic lattice coupled via the nearest neighbour interaction

$$\beta H = -\sum_{\langle ij \rangle} K_{ij} \vec{S}_i \cdot \vec{S}_j. \tag{1}$$

If both sites $i$ and $j$ belong to the surface layer, we set $K_{ij} = K_1$, otherwise, $K_{ij} = K$ (see Fig. 1, left). For $N = 1, 2$ the boundary phase diagram has the schematic shape shown in Fig. 2, left. When the system is tuned to the bulk critical point $K = K_c$ it admits three boundary universality classes:

- the "ordinary" universality class, where the bulk and boundary order at the same bulk coupling,

- the "extraordinary" universality class, where the onset of bulk order occurs in the presence of (quasi) long-range boundary order,

- the "special" universality class, the multicritical point in Fig. 2, left.

The presence of a (quasi)long-range ordered boundary phase for $K < K_c$ and large $K_1/K$ mandates the existence of these three classes.

For $N > 2$, the boundary has a finite correlation length for $K < K_c$. Thus, the existence of a separate extraordinary boundary universality class is not required. Nevertheless, per Ref. [1], the extraordinary boundary universality class actually survives in the range $2 < N < N_c$, where the phase diagram has the shape in Fig. 2, right. Here, $N_c > 2$ is an a priori unknown critical value of $N$. Furthermore, in the range $2 \le N < N_c$ the extraordinary universality class has a boundary spin correlation function that falls off as

$$\langle \vec{S}_x \cdot \vec{S}_y \rangle \sim \frac{1}{(\log|x-y|)^q}, \tag{2}$$

with $q(N)$ a universal power. Thus, in this range of $N$ the extraordinary boundary universality class is labeled as "extraordinary-log". The universal properties of this class, including the power $q$ and the critical value $N_c$, are determined by certain universal amplitudes of the "normal" boundary universality class, where an explicit symmetry breaking field is applied to the boundary. For $N = 3$, recent Monte Carlo simulations find a special fixed point and behavior

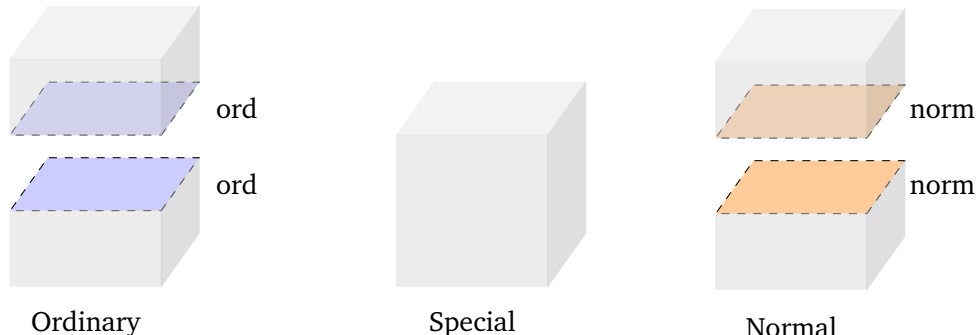

Figure 3: The ordinary, special, and normal fixed points of the 3d O($N$) model with a plane defect. The ordinary fixed point corresponds to two copies of the semi-infinite ordinary (ord) fixed point. The special fixed point corresponds to a bulk with no defect plane. The normal fixed point corresponds to two copies of the semi-infinite normal (norm) fixed point.

at large $K_1$ consistent with the extraordinary-log class [2].[1] This indicates $N_c > 3$. For $N = 2$, the extraordinary-log character of the large $K_1$ region was also verified by Monte Carlo simulations. [5] Furthermore, the normal universality class was recently studied by Monte Carlo in Ref. [6] and the prediction of Ref. [1] for the relation between the extraordinary-log and normal classes was verified for $N = 2, 3$. Finally, Ref. [7] used numerical conformal bootstrap to estimate $N_c \sim 5$. Several scenarios for the evolution of the phase diagram for $N > N_c$ were discussed in Ref. [1]: the simplest possibility is that only the ordinary universality class remains for $N > N_c$ for all values of $K_1$. Part of this paper presents analytical evidence in favour of this simple scenario.

The primary part of the present paper extends the methods of Ref. [1] to the problem of a 2d plane defect[2] embedded in a 3d bulk O($N$) model. As a representative Hamiltonian, we consider Eq. (1) on an infinite cubic lattice, where the nearest neighbour interaction is set to $K_1$ for spins belonging to a plane $z = 0$ and to $K$ otherwise (see Fig. 1, right). While this problem has been considered in the past, [8–10] the precise phase diagram for $N > 2$, in particular the behavior in the region $K_1 > K$, has not been studied carefully.[3] In this paper, we claim that while the phase diagram for $N = 1, 2$ has the expected shape in Fig. 2, left, for all finite $N > 2$ the phase diagram has the shape in Fig. 2, right. In other words, the ordinary, special and extraordinary universality classes all exist for $N \geq 1$. Furthermore, for $N \geq 2$, the extraordinary universality class is of extraordinary-log character, with properties (including the exponent $q$ in Eq. (2)) again determined by those of the normal universality class in a semi-infinite geometry. Thus, unlike in the semi-infinite O($N$) model, there is no critical value $N_c$ above which the extraordinary-log class ceases to exist.

We can argue that the ordinary and extraordinary classes exist for all $N$ for the plane defect as follows. Consider a critical bulk model with no defect, $K_1 = K = K_c$, and then turn on a small $K_1 - K_c$. The resulting continuum action is

$$S = S_{\text{inf}} + c \int d^2 x \, \epsilon(x, z = 0). \tag{3}$$

Here $S_{\text{inf}}$ is the uniform continuum action in the 3d infinite geometry and $\epsilon$ is the relevant

---

[1]Prior Monte Carlo evidence for the existence of a special transition and an extraordinary phase at $N = 3$ had appeared in [3]. See also [4] for a related study at $N = 4$.

[2]We also refer to this as an interface defect.

[3]Another related problem considered in the past is the interface between the free theory and the interacting $O(N)$ model. [11] We do not address this problem in the present manuscript.

bulk $O(N)$ singlet scalar (the so-called "thermal" operator.) The coupling $c$ is proportional to $K_c - K_1$. It is believed that the scaling dimension $\Delta_\epsilon < 2$ for all finite $N \geq 1$. [12,13] Thus, the coupling $c$ is relevant around the $c = 0$ fixed point. All other $O(N)$ singlet defect perturbations are irrelevant.[4] Thus, we have found a special fixed point for all $N$ that simply corresponds to the model with no defect. [8] It is natural to guess that the universality classes on the two sides of the special fixed point $c < 0$ and $c > 0$ are distinct.

For $c > 0$, the model is expected to flow to the ordinary fixed point, which corresponds to the defect plane "cutting" the system into two disconnected halves with each half realizing the semi-infinite ordinary universality class. [8] Indeed, for the semi-infinite ordinary universality class, the most relevant operator is the boundary order parameter $\hat{\phi}_a$ whose dimension is believed to satisfy $\Delta_{\hat{\phi}}^{\text{ord}} > 1$ for all $N$; Monte Carlo simulations for $N = 1, 2, 3$ are consistent with this [3] and large-$N$ calculations give $\Delta_{\hat{\phi}}^{\text{ord}} = 1 + \frac{2}{3N} + O(N^{-2})$. [14] Thus, the action of the ordinary fixed point for the defect plane together with the leading perturbation is

$$S = S_{\text{ord}}^1 + S_{\text{ord}}^2 + u \int \mathrm{d}^2\mathrm{x}\, \hat{\phi}_a^1 \hat{\phi}_a^2. \tag{4}$$

Here $S_{\text{ord}}^{1,2}$ is the action of the semi-infinite ordinary fixed point for each half-space. By the discussion above the perturbation $u$ is irrelevant for all finite $N$, so the ordinary fixed point is stable.

The nature of the extraordinary fixed point realized for $c < 0$ is the main question we address in this paper. Motivated by Ref. [1], we approach this fixed point through study of the normal universality class. For $N = 1$, we expect long range boundary order at the extraordinary fixed point. Due to the rigidity of the Ising order, we expect the extraordinary class to be identical to the normal defect universality class, where an explicit symmetry breaking field is applied on the defect. For all $N$, the normal defect class corresponds to the system cut into two disconnected halves with each half realizing the semi-infinite normal universality class:

$$S = S_{\text{norm}}^1 + S_{\text{norm}}^2. \tag{5}$$

Indeed in the $N = 1$ case, the lowest dimension boundary operator at the semi-infinite normal fixed point is believed to be the displacement D with dimension $\Delta_D = 3$, [15] so the perturbation $\delta L_{bound} \sim D_1 D_2$ coupling the two halves is highly irrelevant. For $N \geq 2$, the lowest dimension boundary operator at the semi-infinite normal fixed point is believed to be the $O(N-1)$ vector $t_i$, with dimension $\Delta_t = 2$, [8] so the coupling $\delta L_{bound} \sim t_i^1 t_i^2$ is again irrelevant.

Starting with this picture of the normal defect universality class for $N \geq 2$, we remove the explicit symmetry breaking boundary field and access the extraordinary universality class using the RG approach of Ref. [1]. We find that an extraordinary-log class is realized for all $N \geq 2$.

Our discussion presently applies to general finite $N$. Further analytical control appears in the large-$N$ limit. When $N = \infty$, the model possesses a line of defect fixed points. [10] Along this line the lowest dimension defect operators are $O(N)$ vectors of even (odd) parity under $z \to -z$ with dimensions $\Delta_S = 1 - \mu$ ($\Delta_A = 1 + \mu$), where $0 \leq \mu \leq 1$ is a coupling constant tuning the system along the line of fixed points. The value $\mu = 0$ corresponds to the ordinary fixed point, $\mu = 1/2$ to the special fixed point (no defect) and $\mu = 1$ to the normal fixed point.[5]

---

[4]The next lowest one is expected to be $\partial_z \epsilon$, which is odd under the reflection symmetry $z \to -z$, thus disallowed in the model (1), but allowed in more general models.

[5]Once $N$ is finite, $\mu$ flows under RG and the approach $\mu \to 1$ gives rise to the extraordinary-log universality class.

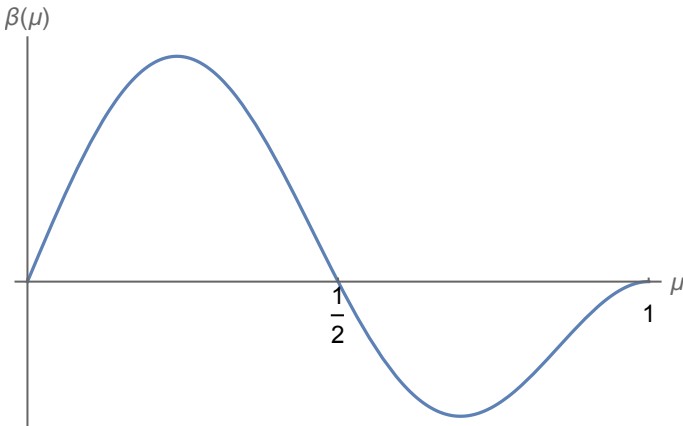

Figure 4: $\beta$-function for the defect coupling constant $\mu$ at $O(1/N)$, Eq. (6). $\mu = 0$ corresponds to the ordinary universality class, $\mu = 1/2$ to the special universality class and the approach $\mu \to 1$ — to the extraordinary-log universality class.

The existence of a line of fixed points is expected to be a peculiarity of the strict $N = \infty$ limit. In this paper, we compute the $\beta$-function for the coupling constant $\mu$ to $O(1/N)$ obtaining:[6]

$$\frac{d\mu}{d\ell} = -\beta(\mu) = \frac{16(\mu^2 - 1/4)}{3N\pi^2} \frac{\sin^2 \mu\pi}{\mu} . \tag{6}$$

Thus, at large finite $N$ the line of fixed points disappears and only the ordinary, special and extraordinary-log universality classes are left, see Fig. 4. The form of the $\beta$-function (6) for $\mu$ close to these fixed points is in agreement with results obtained using other methods. In particular, the behavior of $\beta(\mu)$ for $\mu \to 1$ that controls the extraordinary-log universality class exactly agrees with results obtained using the RG approach of Ref. [1] and provides a non-trivial check of the latter. In addition, the analysis of $\beta(\mu)$ near the special (uniform bulk) fixed point $\mu = 1/2$ confirms that the bulk OPE coefficient $\lambda_{\epsilon\epsilon\epsilon}$ vanishes to $O(1/N^{3/2})$, as found by a direct computation in Ref. [16].

We additionally discuss our results in the context of general theorems for 3d CFTs. It is known that a general conformal boundary of a 3d CFT is characterized by certain "central charges" describing its response to gravity [17–19]:

$$T^\mu_\mu = \frac{\delta(x_\perp)}{24\pi} \left( a\hat{R} + b_K \mathrm{tr}\hat{K}^2 \right) . \tag{7}$$

Here $\hat{R}$ is the boundary Ricci scalar, $\hat{K}$ is the traceless part of the extrinsic curvature associated to the boundary, and $x_\perp$ is the coordinate perpendicular to the boundary. Jensen and O'Bannon in Ref. [17] proved that the coefficient $a$ of the Ricci scalar decreases under boundary RG flow.[7] In particular, $a$ is constant along a line of fixed points. Ref. [22] computed $a$ for the O(N) model with a boundary for both the ordinary and normal fixed points to leading order in $N$. Here we extend their result to first subleading order in $N$:

$$a^O_{bound} = -\frac{1}{16} + O\left(N^{-1}\right), \qquad a^N_{bound} = -\frac{N}{2} - \frac{1}{16} + O\left(N^{-1}\right), \tag{8}$$

where $a^O_{bound}$ ($a^N_{bound}$) stands for the central charge at the ordinary (normal) boundary fixed point. We further consider the central charge for the plane defect. The RG structure of the

---

[6]Here increasing the RG scale $\ell$ corresponds to the flow to the IR.

[7]See also Refs. [20], [21] for alternative proofs.

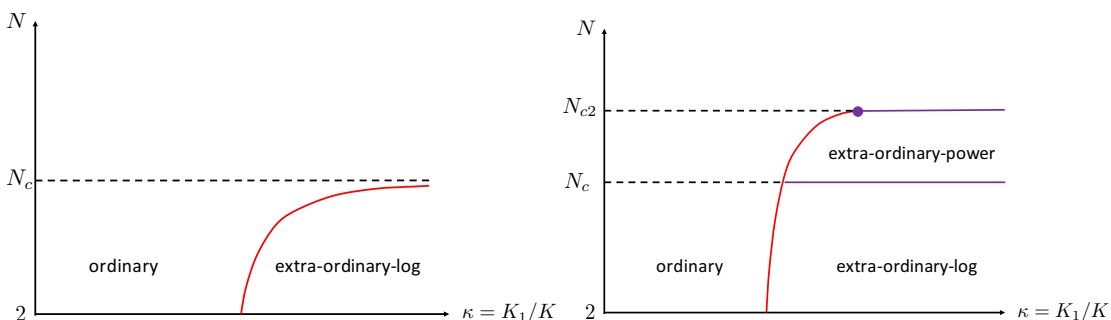

Figure 5: Phase diagram of the semi-infinite classical O($N$) model in 3d with $N \geq 2$ at $K = K_c$ proposed in Ref. [1]. Left: scenario I. Right: scenario II. The dashed lines are a guide to eye and *do not* denote phase transitions. Solid lines are phase transitions. The red curve marks the special transition.

ordinary, extraordinary-log and special interface fixed points implies

$$a_{int}^{O} = 2a_{bound}^{O} = -\frac{1}{8} + O\left(N^{-1}\right), \qquad a_{int}^{eo} = 2a_{bound}^{N} + N - 1 = -\frac{9}{8} + O\left(N^{-1}\right), \qquad a_{int}^{sp} = 0, \tag{9}$$

where the subscript "$int$" denotes the interface central charge and superscripts $O$, $eo$ and $sp$ stand for ordinary, extraordinary and special. The result for the central charge at the special interface fixed point $a_{int}^{sp} = 0$ is exact. In addition, we find by an explicit computation that at $N = \infty$, $a/N = 0$ along the whole line of interface fixed points $0 \leq \mu \leq 1$, consistent with the theorem of Jensen and O'Bannon. Further, to next order in $N$, we find that the differences $a_{int}^{O} - a_{int}^{sp}$ and $a_{int}^{eo} - a_{int}^{sp}$ in Eq. (9) are in agreement with the detailed form of the $a$-theorem of Jensen and O'Bannon, see Eq. (95).

Finally, as already noted, we return to the problem of the O($N$) model in a semi-infinite geometry. In Ref. [1] two scenarios for the evolution of the boundary phase diagram past the critical value $N = N_c$ were proposed. In the first scenario, Fig. 5 (left), the special fixed point approaches the extraordinary fixed point as $N \to N_c^{-}$ and annihilates with it at $N = N_c$, such that only the ordinary universality class remains for $N > N_c$. In the second scenario, Fig. 5 (right), the extraordinary universality class survives for some range $N_c < N < N_{c2}$, where it becomes a true boundary conformal fixed point with a non-trivial scaling dimension $\Delta_{\hat{\phi}} > 0$. This universality class was labeled as "extraordinary-power." Since large-$N$ calculations find only the ordinary universality class in the semi-infinite geometry, the extraordinary-power fixed point would have to annihilate with the special fixed point at some higher critical value of $N = N_{c2}$, so that only the ordinary fixed point would be left for $N > N_{c2}$. The correct of the two scenarios is determined by the sign of a higher order term $b$ in the $\beta$-function of the surface spin-stiffness at $N = N_c$. The computation of $b$ for general $N$ is challenging as it almost certainly requires the knowledge of the four-point function of the tilt operator $\hat{t}_i$ at the normal fixed point. In this paper, we compute $b$ for $N \to \infty$. Assuming that $b(N_c)$ has the same sign as $b(N \to \infty)$, we find that the scenario in Fig. 5 (left) is realized.

This paper is organized as follows. In Sec. 2, we first use RG to show the existence of the extraordinary-log universality class for the 3d O($N$) model with a defect plane for all $N \geq 2$. We then study how the line of defect fixed points present at $N = \infty$ is lifted by $1/N$ corrections in Sec. 3. Next in Sec. 4, we study the boundary and interface central charge $a$. Finally in Sec. 5, we return to the semi-infinite 3d O($N$) model and compute the coefficient $b(N = \infty)$ in the $\beta$-function of the surface spin-stiffness. Some remarks on line defects in 2+1D quantum spin models are made in Sec. 6.

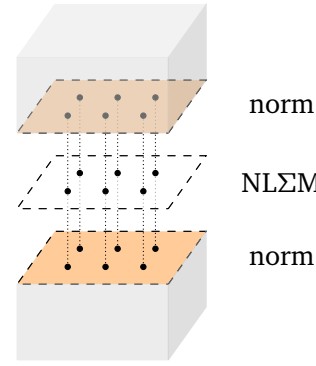

Figure 6: The extraordinary-log fixed point of the 3d O($N$) model with a plane defect. The extraordinary-log fixed point corresponds to two copies of the semi-infinite normal (norm) fixed point, with each boundary coupled to a 2D nonlinear sigma model (NLΣM).

## 2  RG analysis of the plane defect extraordinary fixed point

In this section, we generalize the RG analysis of Ref. [1] to the O($N$) model with a plane defect. We are interested in the large $K_1$ limit of the model (1) where the interface has a strong tendency to local O($N$) order. In this limit, we may describe the defect layer by a non-linear $\sigma$-model for the defect order parameter $\vec{n}$:

$$S_{\vec{n}} = \frac{1}{2g} \int d^2\mathrm{x} \left( \partial_\mu \vec{n} \right)^2 , \qquad \vec{n}^2 = 1 . \tag{10}$$

When $K_1$ is large, the bare coupling $g$ is small and fluctuations of $\vec{n}$ are suppressed at least on short distance scales. Then, $\vec{n}$ acts like a boundary symmetry breaking field for the semi-infinite regions on the two sides of the defect. Thus, there is an intermediate length scale at which the defect is described by the normal fixed point with an additional term that restores O($N$) symmetry at the defect:

$$S = S_{\vec{n}} + S_{\mathrm{norm}}^1 + S_{\mathrm{norm}}^2 - s \int d^2\mathrm{x} \, \pi_i \left( \mathrm{t}_i^1 + \mathrm{t}_i^2 \right) . \tag{11}$$

As in the introduction, $S_{\mathrm{norm}}^1$ and $S_{\mathrm{norm}}^2$ are the actions of the normal fixed points of the semi-infinite regions on each side of the defect. We have also included the leading coupling between the fluctuations of $\vec{n} = (\vec{\pi}, \sqrt{1 - \vec{\pi}^2})$ and the boundary operators of the normal fixed points. Note that we are taking $\vec{n}$ to fluctuate about $\hat{e}_N$, so the symmetry breaking field of the normal fixed points is also along $\hat{e}_N$. The operators $\mathrm{t}_i^1, \mathrm{t}_i^2$, $i = 1 \ldots N-1$ are the "tilt" operators of the normal fixed points, which appear in the boundary OPE of the bulk order parameter as

Normal Fixed Point: 
$$\begin{cases} \phi_N^{1,2}(\mathrm{x}, x^3) \sim \dfrac{a_\sigma}{(2x^3)^{\Delta_\phi}} + b_{\mathrm{D}}(2x^3)^{3-\Delta_\phi} \mathrm{D}^{1,2}(\mathrm{x}) + \ldots, & x^3 \to 0, \quad (12a) \\[2mm] \phi_i^{1,2}(\mathrm{x}, x^3) \sim b_{\mathrm{t}}(2x^3)^{2-\Delta_\phi} \mathrm{t}_i^{1,2}(\mathrm{x}) + \ldots, & x^3 \to 0. \quad (12b) \end{cases}$$

Here $\Delta_\phi$ is the bulk scaling dimension of the order parameter and $\mathrm{D}^{1,2}$ are the displacement operators. All the bulk and boundary operators are normalized as $\langle O^a(x)O^b(y) = \frac{\delta^{ab}}{(x-y)^{2\Delta_O}}$, $\langle \hat{O}^M(x)\hat{O}^N(y) = \frac{\delta^{MN}}{(\mathrm{x}-\mathrm{y})^{2\Delta_{\hat{O}}}}$. The OPE coefficients $a_\sigma$, $b_{\mathrm{t}}$ and $b_{\mathrm{D}}$ are universal constants of the

$$\pi \qquad \mathsf{t}^{1,2} \qquad \pi$$

Figure 7: A contribution to the two-point function of $\pi_i$ from coupling to the tilt operators $\vec{t}_i^{\,1,2}$.

semi-infinite normal fixed point. By the argument applied in Ref. [1] to the semi-infinite system, the parameter $s$ in (11) is fixed by the O($N$) symmetry to be

$$s = \frac{1}{4\pi} \frac{a_\sigma}{b_t} \,. \tag{13}$$

This is exactly the same value of $s$ as in the semi-infinite system. As explained in the introduction, direct coupling between the operators of $S_{\mathrm{norm}}^1$ and $S_{\mathrm{norm}}^2$ is irrelevant. Just as in the semi-infinite geometry, coupling of $\mathsf{t}^{1,2}$ to higher powers of $\vec{\pi}$ is expected to be present and fixed by the O($N$) symmetry in terms of the data of the normal fixed point; such higher order terms won't affect our RG analysis to the leading order in $g$ considered here.

Thus, the coupling $g$ is the only free parameter in the defect action. The perturbative calculation of the $\beta$-function of $g$ proceeds in exactly the same manner as for the semi-infinite system by considering the first correction in $g$ to the one-point function of $n_N$ and the two-point function of $\vec{\pi}$. [1] We obtain:

$$\frac{dg}{d\ell} = -\beta(g) = -\alpha_{\mathrm{plane}}g^2 + O(g^3), \qquad \alpha_{\mathrm{plane}} = \pi s^2 - \frac{N-2}{2\pi} \,. \tag{14}$$

The second term in $\alpha_{\mathrm{plane}}$ gives the standard $\beta$-function of the 2d non-linear $\sigma$-model (10). The first contribution originates from the coupling $s$ in Eq. (11) that enters the two-point function of $\vec{\pi}$ via the diagram in Fig. 7. Note that for a semi-infinite system one has the same form of $\beta(g)$ but with $\alpha_{\mathrm{bound}} = \frac{\pi s^2}{2} - \frac{N-2}{2\pi}$, i.e. the contribution to the $\beta$-function from coupling to the tilt operators is two times larger for the plane defect compared to a semi-infinite system — a straightforward consequence of the coupling to both sides of the plane. This has important physical implications. In the large $N$ limit $a_\sigma$ and $b_t$ have been computed [1] and give:

$$\pi s^2 = \frac{N}{2\pi} + O(1/N), \tag{15}$$

so for the plane defect in the large-$N$ limit

$$\alpha_{\mathrm{plane}} = \frac{1}{\pi} + O(1/N). \tag{16}$$

Thus, for the plane defect $\alpha_{\mathrm{plane}}$ is positive both for $N = 2$ and for $N \to \infty$, suggesting that $\alpha_{\mathrm{plane}}$ remains positive for all $N \geq 2$. Truncated conformal bootstrap estimates of $a_\sigma(N)$ and $b_t(N)$ support this conclusion. [7] Thus, we expect the extraordinary-log class to be realized for all values of $N \geq 2$. Here $g$ flows to zero in the IR as $g^{-1}(\ell) \approx g^{-1} + \alpha_{\mathrm{plane}}\ell$. This is in contrast to the case of a semi-infinite system, where $\alpha_{\mathrm{bound}} = -\frac{N-4}{4\pi} + O(1/N)$ becomes negative for large enough $N$, so the extraordinary-log class is only realized in a finite range $2 \leq N < N_c$. Predictions for $\alpha_{\mathrm{plane}}$ for $N = 2, 3$ based on the Monte-Carlo results for $a_\sigma$ and $b_t$ [6] are given in Table 1.

As for the case of the semi-infinite system, the anomalous dimension of $\vec{n}$, which can be read off from the one-point function of $n_N$ in a symmetry breaking field, is not affected by the coupling $s$ to leading order in $g$:

$$\eta_{\vec{n}}(g) = \frac{(N-1)g}{2\pi} + O(g^2). \tag{17}$$

Table 1: Values of the coefficient $\alpha$ characterizing the extraordinary-log class for $N = 2, 3$ for the semi-infinite ($\alpha_{\text{bound}}$) and plane defect ($\alpha_{\text{plane}}$) geometries obtained from Monte Carlo results for the OPE coefficients $a_\sigma$, $b_t$ of the semi-infinite normal fixed point. [6]

| $N$ | $\alpha_{\text{bound}}$ | $\alpha_{\text{plane}}$ |
|-----|------------|------------|
| 2 | 0.300 (5) | 0.600(10) |
| 3 | 0.190(4) | 0.540(8) |

Here $\eta_{\vec{n}}$ enters the Callan-Symaczik equation for the $m$-point function of $\vec{n}$ as

$$\left( \Lambda \frac{\partial}{\partial \Lambda} + \beta(g)\frac{\partial}{\partial g} + \frac{m}{2}\eta_{\vec{n}}(g) \right) D_{\vec{n}}^m(g, \Lambda) = 0, \tag{18}$$

with $\Lambda$ — the UV cut-off. Integrating the Callan-Symanczik equation for the two-point function, we obtain

$$\langle \vec{n}(x) \cdot \vec{n}(0) \rangle \propto \frac{1}{(\log x)^q}, \qquad x \to \infty, \tag{19}$$

with

$$q = \frac{N-1}{2\pi\alpha}. \tag{20}$$

## 3 The plane defect in the large-$N$ expansion

Now that we have given evidence for the existence of the extraordinary-log fixed point for $2 \le N < \infty$, we show how the ordinary, special and extraordinary-log fixed points are recovered at large $N$. Recall that the bulk continuum action for the O($N$) model is

$$S_{\text{inf}} = \int \mathrm{d}^3x \left[ \frac{1}{2}(\partial_\mu \vec{\phi})^2 + \frac{i\lambda}{2}\left( \vec{\phi}^2 - \frac{1}{g_{\text{bulk}}} \right) \right], \tag{21}$$

where $\vec{\phi}$ is the continuum O($N$) field, $i\lambda$ is a Lagrange multiplier that fixes the norm of $\vec{\phi}$, and the coupling $g_{\text{bulk}}$ is tuned to the critical point. In the presence of a plane defect at $x^3 = 0$, we label fields on either side of the plane defect $\vec{\phi}^1$ and $\vec{\phi}^2$, as well as $\lambda^1, \lambda^2$. Then, the bulk action can be written as

$$S_{\text{inf}} = \int_{x^3 \geq 0} \mathrm{d}^3x \sum_{m=1,2} \left[ \frac{1}{2}(\partial_\mu \vec{\phi}^m)^2 + \frac{i\lambda^m}{2}\left( (\vec{\phi}^m)^2 - \frac{1}{g_{\text{bulk}}} \right) \right]. \tag{22}$$

Here, we label the coordinates $x = (\mathbf{x}, x^3)$, where the last component corresponds to the direction normal to the plane defect.

At $N = \infty$, we need to solve the saddle-point equation for $\langle i\lambda \rangle$ and the $\phi$ propagator, $\langle \phi_a^m(x)\phi_b^n(x') \rangle = \delta_{ab} G^{mn}(x, x')$, $m, n = 1, 2$:

$$\left( -\partial_x^2 + \langle i\lambda(x) \rangle \right) G^{mn}(x, x') = \delta^{mn}\delta^3(x - x'), \quad G^{11}(x, x) = G^{22}(x, x) = \frac{1}{Ng_c}. \tag{23}$$

The last condition can be understood from the bulk OPE, $\sum_a \phi^a \times \phi^a \sim 1 + i\lambda + \ldots$, from which

$$G^{11}(x, y) = G^{22}(x, y) = \frac{1}{4\pi|x - y|} + O(|x - y|). \tag{24}$$

Conformal invariance dictates that $i\lambda$, a field with dimension 2, acquires an expectation value parametrized by a coupling constant $\mu$:

$$\langle i\lambda(x)\rangle = \frac{\mu^2 - 1/4}{(x^3)^2}\,.\tag{25}$$

Similarly, conformal invariance fixes

$$\left\langle \phi_a^1(x)\phi_b^1(x')\right\rangle = \delta_{ab}\frac{g_{11}(v)}{\sqrt{x^3 x'^3}}\,,\qquad \left\langle \phi_a^1(x)\phi_b^2(x')\right\rangle = \delta_{ab}\frac{g_{12}(v)}{\sqrt{x^3 x'^3}}\,,\tag{26}$$

$$v = \frac{(x^3)^2 + (x'^3)^2 + r^2}{2x^3 x'^3}\,,\qquad r = |\mathbf{x}-\mathbf{x}'|\,.\tag{27}$$

Then Eq. (23) implies $\mathcal{D}g^{11}(v) = \mathcal{D}g^{12}(v) = 0$, apart from contact terms at $v \to 1$. Here

$$\mathcal{D} = \mu^2 - 1 - 3v\frac{\mathrm{d}}{\mathrm{d}v} - (v^2 - 1)\frac{\mathrm{d}^2}{\mathrm{d}v^2}\,.\tag{28}$$

There are two linearly independent solutions of the equation $\mathcal{D}g(v) = 0$,

$$s_{S/A}(v) = \frac{(v + \sqrt{v^2 - 1})^{\pm\mu}}{\sqrt{v^2 - 1}}\,,\tag{29}$$

and $g_{11}(v)$, $g_{12}(v)$ are particular linear combinations of these:

$$g_{11}(v) = \frac{s_S(v) + s_A(v)}{8\pi}\,,\qquad g_{12}(v) = \frac{s_S(v) - s_A(v)}{8\pi}\,.\tag{30}$$

$g_{11}$ is fixed by the OPE (24). $g_{12}$ is fixed by i) the requirement that it be non-singular as $v \to 1$ (since $\phi^1$ and $\phi^2$ live on opposite sides of the defect, their OPE is nonsingular); ii) the requirement that $g_{11}$, $g_{12}$ have the same asymptotic in the boundary limit $v \to \infty$, i.e. the bulk to boundary OPE of $\phi^1$ and of $\phi^2$ is dominated by the same operator.

Given these solutions, we require that $\mu$ be real, in which case, without loss of generality it can be chosen to be positive. Further, $\mu < 1$ so that $g^{11}$, $g^{12}$ go to zero for large $v$, i.e. the $O(N)$ symmetry is not broken and clustering is obeyed. Defining symmetric/anti-symmetric fields from the two $\vec{\phi}^m$ fields is convenient for the rest of the paper:

$$\phi_a^{S/A} = \frac{1}{\sqrt{2}}(\phi_a^1 \pm \phi_a^2)\,.\tag{31}$$

Then,

$$\left\langle \phi_a^S(x)\phi_b^S(x')\right\rangle = \frac{\delta_{ab}s_S(v)}{4\pi\sqrt{x^3 x'^3}}\,,\qquad \left\langle \phi_a^A(x)\phi_b^A(x')\right\rangle = \frac{\delta_{ab}s_A(v)}{4\pi\sqrt{x^3 x'^3}}\,,\qquad \left\langle \phi_a^S(x)\phi_b^A(x')\right\rangle = 0\,.\tag{32}$$

Thus, at $N = \infty$ we find a line of boundary fixed points parameterized by $0 \le \mu \le 1$. We recall that for a (normalized) bulk scalar conformal primary $O(x)$ of dimension $\Delta_O$,

$$\langle O(x)O(x')\rangle = \frac{1}{(4x^3 x'^3)^{\Delta_O}}\sum_n b_n^2 f_{bry}(\hat{\Delta}_n, v)\,,\tag{33}$$

where the sum runs over the operators appearing in the bulk to boundary OPE of $O(x)$ with $\hat{\Delta}_n$ - the boundary operator scaling dimension and $b_n$ – the OPE coefficient. In spacetime dimension $D = 3$,

$$f_{bry}(\hat{\Delta}, v) = 2^{2\hat{\Delta}-1}\frac{(v + \sqrt{v^2 - 1})^{1-\hat{\Delta}}}{\sqrt{v^2 - 1}}\,.\tag{34}$$

Thus, we see that at this order in $N$, the bulk to boundary OPE of $\phi^S$ ($\phi^A$) is saturated by a single boundary operator with dimension $\hat{\Delta}_S = 1 - \mu$, ($\hat{\Delta}_A = 1 + \mu$).

The ordinary, special, and normal fixed points are all visible in the range $\mu \in [0, 1]$. At the ordinary fixed point $\mu = 0$, the plane defect action is equivalent to two copies of the half-space action (evident in Eq. (4) for $u = 0$). Thus, as expected, the propagators at $\mu = 0$ match the $N = \infty$ result from Ref. [14] for the ordinary fixed point for an O($N$) model on a 3D half-space. At the special fixed point $\mu = 1/2$, there is effectively no defect plane, the model is translationally invariant, and the propagators, as expected, take the form

$$\left\langle \phi_a^1(x) \phi_b^{1/2}(x') \right\rangle_{\mathrm{sp}} = \frac{\delta_{ab}}{4\pi \sqrt{|\mathbf{x}' - \mathbf{x}|^2 + (x'^3 \mp x^3)^2}} \,. \tag{35}$$

Finally, at the normal fixed point $\mu = 1$, the plane defect action is again equivalent to two copies of the half-space normal action. Indeed, for a half-space normal fixed point with the magnetic field pointing along the $N$th direction, we have [1]

$$\langle \phi_N(x) \rangle_{\mathrm{norm}}^2 = \frac{N}{4\pi x^3}, \qquad \langle \phi_N(x) \phi_N(x') \rangle_{\mathrm{norm,c}} \sim O(N^0),$$

$$\langle \phi_i(x) \phi_j(x') \rangle_{\mathrm{norm}} = \frac{\delta_{ij}}{4\pi (x^3 x'^3)^{1/2}} \left( \frac{v}{\sqrt{v^2 - 1}} - 1 \right), \qquad i, j = 1 \ldots N \,. \tag{36}$$

Here and below the subscript c stands for the connected part of the two-point function. Then the O($N$) invariant combinations $\sum_{a=1}^N \langle \phi_a^1(x) \phi_a^{1/2}(x') \rangle_{\mathrm{norm}}$ for two decoupled normal half-space actions exactly match Eqs. (29), (30) with $\mu \to 1$.

## 3.1 The $\lambda$ Propagator

We now study the plane defect for large but finite $N$. We specifically compute the renormalization group (RG) flow for the coupling constant $\mu$ using the $1/N$ correction to $\langle i\lambda(x) \rangle$. To compute the RG flow, we first compute $\langle \lambda(x) \lambda(x') \rangle_c$, which is nonzero only to order $1/N$.

Recall that the bulk partition function is

$$\mathcal{Z}_{\mathrm{bulk}} = \int \mathcal{D}\vec{\phi} \mathcal{D}\lambda \exp\left\{ -\int \mathrm{d}^3 x \left[ \frac{1}{2} (\partial_\mu \vec{\phi})^2 + \frac{i\lambda}{2} \left( \vec{\phi}^2 - \frac{1}{g_{\mathrm{bulk}}} \right) \right] \right\}. \tag{37}$$

We now add

$$-\int \mathrm{d}^3 x \, \mathrm{d}^3 x' \frac{1}{2} \lambda(x) K(x, x') \lambda(x') + \int \mathrm{d}^3 x \, \mathrm{d}^3 x' \frac{1}{2} \lambda(x) K(x, x') \lambda(x') \tag{38}$$

to the action, such that upon integrating out the $\phi$ fields, the second order term in $\lambda$ in the original action cancels with the first new term [23]. Then,

$$K(x, x') = \frac{N G(x, x')^2}{2}, \qquad \int \langle \lambda(x) \lambda(y) \rangle_c K(y, z) \, \mathrm{d}^3 y = \delta^3(x - z). \tag{39}$$

The method for finding a solution to Eq. (39) is explained in Ref. [24]. We detail the specific computation in App. A and present the results of the computation here for both sides of the defect plane:

$$\left\langle \lambda^1(x) \lambda^1(x') \right\rangle_c = \frac{2}{(4x^3 x'^3)^2 N} h_{11}(v), \quad h_{11}(v) = \frac{32 \cos^2(\mu\pi)}{(v+1)^2 \pi^2} - \frac{32}{(v-1)^2 \pi^2}, \tag{40}$$

$$\left\langle \lambda^1(x) \lambda^2(x') \right\rangle_c = \frac{2}{(4x^3 x'^3)^2 N} h_{12}(v), \quad h_{12}(v) = -\frac{32 \sin^2(\mu\pi)}{(v+1)^2 \pi^2} \,. \tag{41}$$

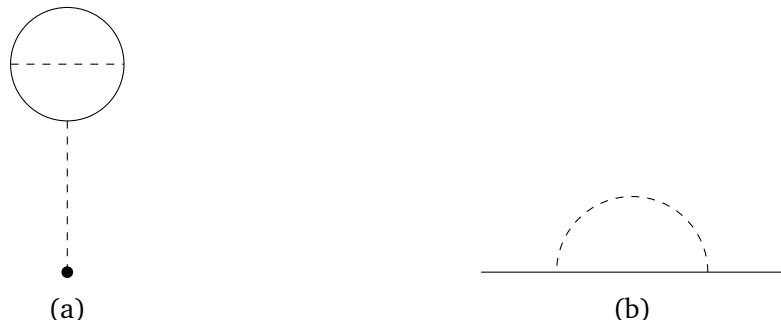

Figure 8: (a): The diagram for the order $1/N$ correction to $\langle i\lambda \rangle$.
(b): The diagram required for computing the bubble in the left diagram. In each diagram, the dashed line is the $\lambda$ propagator, the solid line is the $\phi$ propagator, and the solid vertex inserts $i\lambda$.

As expected, the two-point function of $\lambda$ at $\mu = 0$ and $\mu = 1$ matches the result for the ordinary, [14] and normal fixed point, [25] while at $\mu = 1/2$ we recover the two-point function without the plane defect.

We similarly define symmetric and anti-symmetric analogues of $\lambda$:

$$\lambda^{S/A}(x) = \frac{1}{\sqrt{2}}\left(\lambda^1(x) \pm \lambda^2(x)\right). \tag{42}$$

Then,

$$\left\langle \lambda^S(x)\lambda^S(x')\right\rangle_c = \frac{2}{(4x^3 x'^3)^2 N}h_S(v), \quad h_S(v) = \frac{32\cos(2\mu\pi)}{(v+1)^2\pi^2} - \frac{32}{(v-1)^2\pi^2}, \tag{43}$$

$$\left\langle \lambda^A(x)\lambda^A(x')\right\rangle = \frac{2}{(4x^3 x'^3)^2 N}h_A(v), \quad h_A(v) = \frac{32}{(v+1)^2\pi^2} - \frac{32}{(v-1)^2\pi^2}. \tag{44}$$

Expanding these two-point functions in boundary conformal blocks (33), (34), we find operators with dimension $\hat{\Delta} = 2, 3, 4, 5, \dots$ in the bulk to boundary OPE of $\lambda^S$ and operators with dimension $\hat{\Delta} = 3, 5, 7, 9, \dots$ in the bulk to boundary OPE of $\lambda^A$.[8] The operator with $\hat{\Delta} = 2$ in the $\lambda_S$ OPE is the marginal operator that tunes the boundary along the line of fixed points parameterized by $\mu$, while the operators with $\hat{\Delta} = 3$ in the $\lambda_S$ and $\lambda_A$ OPEs correspond to the symmetric/antisymmetric combinations of displacement operators.

## 3.2 Renormalization Group Flow for $\mu$

We now compute the renormalization group flow for $\mu$ via the Callan-Symanzik equation for $\langle i\lambda(x)\rangle$. The diagrams required for computing $\langle i\lambda(x)\rangle$ to order $1/N$ are shown in Fig. 8. We evaluate the diagram in Fig. 8(b) at coincident points, after subtracting off bulk divergences, to compute the bubble in diagram Fig. 8(a).

We detail the evaluation of Fig. 8(b) at coincident points in App. B and present the results here. The full form of Fig. 8(b) has both a conformal and nonconformal component. The conformal component, at coincident points, evaluates to

$$G^{11,(b)}_{\text{conf., sub.}}(x,x) = -\frac{2}{3\pi N x^3}(\mu^2 - 1/4) \tag{45}$$

---

[8] Of course, the boundary identity operator is also present in the bulk to boundary OPE of $\lambda_S$, see Eq. (25).

after subtracting off bulk divergences. Then, the $1/N$ contribution to $\langle i\lambda(0,z)\rangle$ is

$$\delta_{\text{conf}}\Big\langle i\lambda\big(0,x'^3\big)\Big\rangle = \frac{1}{2}\int_{\mathbb{R}^{3+}}\mathrm{d}^2r\,\mathrm{d}x^3$$
$$\times\Big[\big\langle\lambda^1\big(0,x'^3\big)\lambda^1\big(\mathbf{r},x^3\big)\big\rangle_{\text{c}}+\big\langle\lambda^1\big(0,x'^3\big)\lambda^2\big(\mathbf{r},x^3\big)\big\rangle_{\text{c}}\Big]G_{\text{conf., sub.}}^{11,(b)}(x,x).\tag{46}$$

All integrals here and below are over half-space. Following the methods of Ref. [1], this integral, to logarithmic accuracy, simplifies to

$$\delta_{\text{conf}}\Big\langle i\lambda\big(0,x'^3\big)\Big\rangle = \frac{16(\mu^2-1/4)}{3N\pi^2}\frac{\mathrm{d}}{\mathrm{d}z}\int_0^\infty\mathrm{d}x^3\left[\frac{\cos(2\mu\pi)}{x'^3+x^3}-\frac{P}{x'^3-x^3}\right]\frac{1}{x^3}$$
$$= \frac{32\sin^2(\mu\pi)(\mu^2-1/4)}{3N\pi^2(x'^3)^2}\log\big(\Lambda x'^3\big),\tag{47}$$

where $P$ denotes principal value, and $1/\Lambda$ is a lattice cutoff.

The full nonconformal component of Fig. 8(b) is

$$G_{\text{nconf}}^{11,(b)}(x,y) = \frac{32(\mu^2-1/4)}{3N\pi^2}\int\mathrm{d}^3w\,\frac{\log\big(\Lambda'w^3\big)}{(w^3)^2}\big(G^{11}(x,w)G^{11}(w,y)+G^{12}(x,w)G^{12}(w,y)\big)$$
$$-\frac{4}{3N\pi^2}\log\big(4x^3y^3\Lambda''^2\big)G^{11}(x,y).\tag{48}$$

Here, $\Lambda'$ and $\Lambda''$ are two UV cutoffs that are lattice-dependent (they are not necessarily equal, but they both inversely scale with the lattice spacing, as does $\Lambda$). Then, the contribution from this term to $\langle i\lambda(0,z)\rangle$ is

$$\delta_{\text{nconf}}\Big\langle i\lambda\big(0,x'^3\big)\Big\rangle = \frac{16(\mu^2-1/4)}{3\pi^2}\int\mathrm{d}^3w\,\mathrm{d}^3x\,\frac{\log\big(\Lambda'w^3\big)}{(w^3)^2}\big(G^{11}(x,w)^2+G^{12}(x,w)^2\big)$$
$$\times\Big[\big\langle\lambda^1(0,x'^3)\lambda^1(x)\big\rangle_{\text{c}}+\big\langle\lambda^1(0,x'^3)\lambda^2(x)\big\rangle_{\text{c}}\Big].\tag{49}$$

Note that we drop the contribution from the term proportional to $\log\big(x^3\Lambda''\big)G^{11}(x,x)$ because together with the subtraction implicit in (45) it contributes to a shift of the critical value of $g_{\text{bulk}}$. Using that the $\lambda$ propagator is, up to a constant, the inverse of the squared $\phi$ propagator, Eq. (39), we obtain a contribution

$$\delta_{\text{nconf}}\Big\langle i\lambda(0,x'^3)\Big\rangle = \frac{32(\mu^2-1/4)}{3N\pi^2}\frac{\log\big(\Lambda'x'^3\big)}{(x'^3)^2}.\tag{50}$$

Combining Eqs. (47), (50), we obtain to logarithmic accuracy

$$\langle i\lambda(0,z)\rangle = \frac{\mu^2-1/4}{z^2}\left(1+\frac{32}{3N\pi^2}(\sin^2(\mu\pi)+1)\log\big(\Lambda x'^3\big)\right).\tag{51}$$

Per the Callan-Symanzik equation,

$$\left(\beta(\mu)\frac{\mathrm{d}}{\mathrm{d}\mu}+\Lambda\frac{\mathrm{d}}{\mathrm{d}\Lambda}+\gamma_\lambda\right)\Big\langle i\lambda(0,x'^3)\Big\rangle = 0,\tag{52}$$

where $\beta(\mu)=-\frac{d\mu}{d\ell}$ is the beta function, or RG flow, of $\mu$, and $\gamma_\lambda\approx-\frac{32}{3\pi^2 N}$ is the anomalous dimension of $\lambda$. We thus find

$$\beta(\mu) = -\frac{16(\mu^2-1/4)}{3N\pi^2}\frac{\sin^2(\mu\pi)}{\mu}.\tag{53}$$

A plot of the beta function is shown in Fig. 4. Thus, for large but finite $N$, we indeed have three fixed points corresponding to the normal, special, and extraordinary-log phases, with the special fixed point unstable and the other two fixed points stable in the IR.

### 3.3 Renormalization Group Flow Near Fixed Points

As we explain below, the RG flow of $\mu$ near the ordinary, special, and normal fixed points for the plane defect system confirms nontrivial results in the literature for the $O(N)$ model with different boundary conditions. Most importantly, $\beta(\mu)$ near the normal fixed point agrees with the RG treatment of section 2.

Let us begin near the normal fixed point, $\mu \to 1$. From (53), we have

$$\beta(\mu) \approx \frac{4}{N}\left(-(1-\mu)^2 + \frac{5}{3}(1-\mu)^3\right) + \dots, \qquad \mu \to 1. \tag{54}$$

We relate $1 - \mu$ to the coupling constant $g$ in (10) by matching the scaling dimension $\hat{\Delta}_S = 1 - \mu = \frac{\eta_{\bar{n}}(g)}{2}$ and arriving at $1 - \mu = \frac{Ng}{4\pi} + O(g^2)$. Note that we've kept only the leading order term in $N$. From this,

$$\beta(g) \approx \frac{g^2}{\pi} - \frac{5}{12\pi^2}Ng^3 + O\left(g^4\right), \qquad N \to \infty. \tag{55}$$

The leading $O(g^2)$ term agrees with (16). Note that the coefficient of the $O(g^3)$ term is insensitive to the re-parameterization $g \to g + O(g^2)$, and thus can be extracted reliably in the $N \to \infty$ limit.

Next, we discuss the special fixed point, $\mu \to 1/2$. We would like to compare our results to the treatment based on the action (3). We take $\epsilon(x)$ to be normalized $\langle \epsilon(x)\epsilon(y) \rangle = \frac{1}{(x-y)^{2\Delta_\epsilon}}$, $\Delta_\epsilon = 2 - \frac{32}{3\pi^2 N} + O(N^{-2})$. Using the OPE

$$\epsilon(x)\epsilon(0) = \frac{1}{x^{2\Delta_\epsilon}} + \frac{\lambda_{\epsilon\epsilon\epsilon}}{x^{\Delta_\epsilon}}\epsilon(0) + \dots, \tag{56}$$

we obtain the RG flow of coupling $c = \Lambda^{2-\Delta_\epsilon}\tilde{c}$ in (3):

$$\beta(\tilde{c}) = -(2 - \Delta_\epsilon)\tilde{c} + \pi\lambda_{\epsilon\epsilon\epsilon}\tilde{c}^2 + O\left(\tilde{c}^3\right), \qquad N \to \infty. \tag{57}$$

While in general dimension $D$ the coefficient $\lambda_{\epsilon\epsilon\epsilon} \sim O(N^{-1/2})$, it has been known for some time that in $D = 3$ the leading $N$ term in $\lambda_{\epsilon\epsilon\epsilon}$ vanishes. [26] Actually, a recent calculation of Ref. [16] shows that for $D = 3$ the first subleading term in $N$ vanishes as well, so $\lambda_{\epsilon\epsilon\epsilon} \sim O(N^{-5/2})$. We verify this result here by comparing (57) to Eq. (53),

$$\beta(\mu) \approx \frac{32}{3N\pi^2}\left(-(\mu - 1/2) + (\mu - 1/2)^2 + O\left((\mu - 1/2)^3\right)\right), \qquad N \to \infty. \tag{58}$$

We need the relation between $\mu$ and $\tilde{c}$. To leading order in $N$, this can be read-off by computing $\langle \epsilon(x) \rangle$ using perturbation theory in $c$ and relating it to $\langle i\lambda(x) \rangle$, (25). We have

$$\langle \epsilon(x) \rangle = -c \int d^2y \, \langle \epsilon(x)\epsilon(y,0)\rangle_{bulk} + \frac{c^2}{2}\int d^2y d^2z \, \langle \epsilon(x)\epsilon(y,0)\epsilon(z,0)\rangle_{bulk} + O\left(c^3\right). \tag{59}$$

Using the normalization of bulk $\lambda$ two-point function (43), to leading order in $1/N$, $i\lambda(x) = \frac{4\Lambda^{2-\Delta_\epsilon}}{\pi\sqrt{N}}\epsilon(x)$, where we introduce a power of the cut-off $\Lambda$ to make dimensions match. Then performing the first integral in (59),

$$\tilde{c} = -\frac{\sqrt{N}}{4}(\mu - 1/2) + O(\mu - 1/2)^2, \qquad N \to \infty, \tag{60}$$

and the leading (linear) terms in $\beta(\mu)$ and $\beta(\tilde{c})$ match. To compare subleading (quadratic) terms, we need a relation between $\mu - 1/2$ and $\tilde{c}$ to quadratic order. We have

$$\langle \epsilon(x)\epsilon(y)\epsilon(z) \rangle_{bulk} = \frac{\lambda_{\epsilon\epsilon\epsilon}}{(x-y)^{\Delta_\lambda}(y-z)^{\Delta_\lambda}(z-x)^{\Delta_\lambda}}. \tag{61}$$

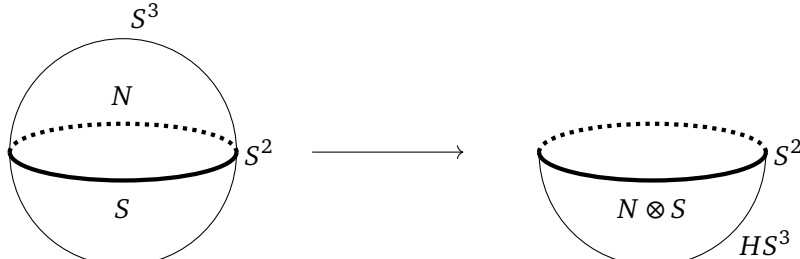

Figure 9: A 2D projection of the folding trick. Instead of considering one field that lives on $S^3$, we consider two fields that live on $HS^3$ and are coupled at the boundary $S^2$.

Using the old result [26], $\lambda_{\epsilon\epsilon\epsilon} \sim O(N^{-3/2})$, the $O(c^2)$ term in (59) is suppressed by $1/N$ compared to the $O(c)$ term. Thus, matching to (25),

$$\tilde{c} = -\frac{\sqrt{N}}{4}\left((\mu-1/2)+(\mu-1/2)^2+O((\mu-1/2)^3)\right), \quad N \to \infty. \tag{62}$$

Then comparing (57), (58), we conclude that $\lambda_{\epsilon\epsilon\epsilon} = 0$ to $O(N^{-3/2})$, in agreement with Ref. [16]. We note that the calculation of $\lambda_{\epsilon\epsilon\epsilon}$ to $O(N^{-3/2})$ in [16] involved multi-loop diagrams, whereas here we reproduce their result with just a one-loop calculation in the presence of a plane defect.

We finally discuss the ordinary fixed point $\mu \to 0$. Here, Eq. (53) gives

$$\beta(\mu) \approx \frac{4}{3N}\mu + O(\mu^3). \tag{63}$$

We would like to connect this result to the treatment (4). We have

$$\beta(u) = 2(\Delta_{\text{ord}} - 1)u + O(u^3), \tag{64}$$

where $\Delta_{\text{ord}} = 1 + \frac{2}{3N} + O(N^{-2})$. [14] This agrees with $\beta(\mu)$ provided that $\frac{d\mu}{du}$ is finite for $u \to 0$. In appendix C, we verify this fact.

# 4 Boundary and interface central charge

In this section, we study the central charge $a$ in Eq. (7) for the boundary and interface defects. This section is structured as follows. In section 4.1 we explicitly show that at $N = \infty$ the central charge $a_{int}/N = 0$ along the entire line of interface fixed points $0 \le \mu \le 1$, in agreement with the $a$-theorem of Jensen and O'Bannon. [17] In sections 4.2 and 4.3 we compute the central charge for the ordinary and normal boundary fixed points to $O(N^{-1})$ obtaining the result (8). This immediately yields the interface central charge for the ordinary and extraordinary-log interface fixed points (9). Finally, in section 4.4 we compare the result for the interface central charge (9) to a detailed form of the $a$-theorem of Jensen and O'Bannon relating the difference of $a$ between the IR and UV fixed points to a particular two point function of the stress-energy tensor, Eq. (95). This gives a highly non-trivial check of the details of the RG flow from the special to ordinary/extraordinary-log interface fixed points at large finite $N$, and of the full $\beta$-function (53) in particular.

## 4.1 Interface central charge at $N = \infty$

We first verify explicitly that at $N = \infty$, $a_{int}/N = 0$ along the entire line of interface fixed points as expected by the monotonicity theorem in [17]. We extract the coefficient $a_{int}$ from

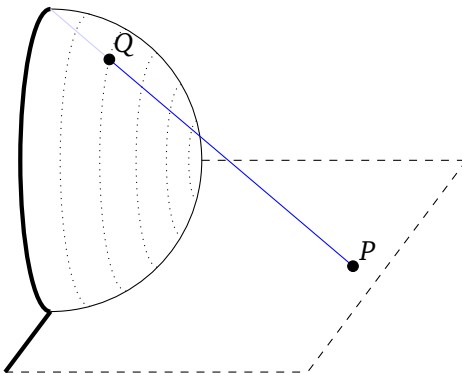

Figure 10: A 2D depiction of the stereographic projection of $HS^3$ onto $\mathbb{R}^3_+$. The boundary $S^2$ is mapped to the plane $x^3 = 0$. Any point $P$ in $\mathbb{R}^3_+$ is mapped to the point $Q$ on $HS^3$ that lies on the line segment connecting $P$ to the north pole, as depicted by the blue line in the figure.

the free energy of the system on a sphere $S^3$ with the defect along its equator $S^2$ [17] (see Fig. 9, left):

$$F_{S^3,int} = -\log Z = -\frac{a_{int}}{3} \log R/\epsilon \,, \tag{65}$$

where $R$ is the radius of the sphere and $\epsilon$ is a UV cut-off. Equivalently, we can use the "folding trick" to think of the system as a "doubled" theory on a hemisphere $HS^3$, where the two copies of the theory are decoupled in the bulk, but generally coupled on the boundary (see Fig. 9, right).

We begin by pointing out that for the special fixed point, $a^{sp}_{int} = 0$ for *any N*.[9] Indeed, the special fixed point corresponds to a trivial interface. Thus, in the unfolded picture we simply have the $O(N)$ model on the sphere $S^3$ with no defect. The partition function of a CFT on $S^3$ is a universal number independent of the sphere radius $R$. Thus, we conclude $a^{sp}_{int} = 0$. Then by the theorem of [17], at finite $N$, $a^O_{bound} < 0$ for the ordinary boundary fixed point (i.e. for a single copy of the $O(N)$ model). Indeed, in the interface model, there is a flow from the special to the ordinary fixed point, and the interface ordinary fixed point is equivalent to two decoupled boundary ordinary fixed points for each side of the interface (see discussion around Eq. (4)).

We now proceed to the explicit computation of the sphere with defect free energy at $N = \infty$. Our calculation follows Refs. [22, 27]. We consider the action

$$S = \frac{1}{2} \int_{HS^3} d^D x \sqrt{g} \sum_{m=1}^{2} \left[ \partial_\mu \phi^m_a g^{\mu\nu} \partial_\nu \phi^m_a + i\lambda^m \left( \phi^m_a \phi^m_a - \frac{1}{g} \right) + \frac{(D-2)}{4(D-1)} \mathcal{R} \phi^m_a \phi^m_a \right]. \tag{66}$$

We work in the "folded" picture: the theory lives on a hemisphere of radius $R$, the index $m$ runs over two copies of the $O(N)$ model, $g$ is the metric, and $\mathcal{R}$ is the Ricci scalar. We've added the conformal coupling to curvature (which ensures that $i\lambda$ transforms as a conformal primary for $N = \infty$). The metric is given by

$$ds^2_{HS^3} = R^2 \left( d\alpha^2 + \sin^2 \alpha \, d\Omega_2^2 \right). \tag{67}$$

Here $d\Omega_2^2 = d\theta^2 + \sin^2 \theta d\varphi^2$ is the metric of a two-sphere, with $\theta$ and $\varphi$ the usual polar coordinates, and $\alpha \in [0, \pi/2]$. $\alpha = \pi/2$ gives the boundary of $HS^3$, which is just $S^2$. This

---

[9]We thank Yifan Wang for pointing out the argument below.

metric is conformally equivalent to flat semi-infinite space, parametrized as $(x^1, x^2, x^3)$ with $x^3 \geq 0$. Indeed, let

$$x^1 = \frac{\sin\alpha\sin\theta\cos\varphi}{1 - \sin\alpha\cos\theta}, \qquad x^2 = \frac{\sin\alpha\sin\theta\sin\varphi}{1 - \sin\alpha\cos\theta}, \qquad x^3 = \frac{\cos\alpha}{1 - \sin\alpha\cos\theta}. \tag{68}$$

This is just the stereographic projection of $S^3$ onto $\mathbb{R}^3$, with the half-sphere $HS^3$ mapping to the half-space $x^3 \geq 0$, which we label $\mathbb{R}^3_+$ (see Fig. 10). The boundary of $HS^3$ maps to the $x^3 = 0$ plane plus the point at infinity. The metric thus is

$$ds^2_{HS^3} = \Omega^2(x)\sum_i (dx_i^2), \quad \Omega(x) = R\frac{2}{(x^1)^2 + (x^2)^2 + (x^3)^2 + 1} = R(1 - \sin\alpha\cos\theta). \tag{69}$$

Now in the semi-infinite geometry $\langle i\lambda^m(x)\rangle_{\mathbb{R}^3_+} = \frac{\mu^2 - 1/4}{(x^3)^2}$, see Eq. (25). Therefore, performing a Weyl transformation yields

$$\langle i\lambda^m(x)\rangle_{HS^3} = \Omega^{-\Delta_\lambda}(x)\langle i\lambda^m(x)\rangle_{\mathbb{R}^3_+} = R^{-2}(\mu^2 - 1/4)\sec^2\alpha, \tag{70}$$

where we used $\Delta_\lambda = 2$ for $N = \infty$. Since $\langle i\lambda^m\rangle$ is $m$ independent, we simply denote it by $\langle i\lambda\rangle$ below. We perform a transformation to symmetric and anti-symmetric components of $\phi$, see Eq. (31). Then at $N = \infty$,

$$F_{S^3,int} = \frac{N}{2}\left[\mathrm{Tr}_S\log\left(-\Delta + \langle i\lambda\rangle + \frac{3}{4R^2}\right) + \mathrm{Tr}_A\log\left(-\Delta + \langle i\lambda\rangle + \frac{3}{4R^2}\right)\right]. \tag{71}$$

The subscripts $S/A$ on the trace indicate that the trace should be performed over eigenstates with boundary conditions appropriate to $\phi^S$ and $\phi^A$ respectively. The constant of $\frac{3}{4R^2}$ in brackets comes from the conformal coupling ($\mathcal{R} = \frac{D(D-1)}{R^2}$ on a $D$ sphere of radius $R$). In appendix D, we repeat the calculation of the trace in Ref. [27] to obtain to logarithmic accuracy:

$$\frac{1}{2}\mathrm{Tr}_{S/A}\log\left(-\Delta + \langle i\lambda\rangle + \frac{3}{4R^2}\right) = \mp\frac{1}{6}\mu^3\log R, \tag{72}$$

where the $+$ sign corresponds to $\phi_A$ (boundary field exponent $\hat{\Delta} = 1 + \mu$) and the $-$ sign to $\phi_S$ ($\hat{\Delta} = 1 - \mu$). This agrees, as expected, with the result of Ref. [22] for the free energy of a free scalar of mass $m^2 = \mu^2 - \frac{1}{4}$ on $AdS_3$ with a spherical boundary. (Indeed, $AdS_3$ is conformally equivalent to $HS^3$ and $\langle i\lambda(x)\rangle_{HS^3}$ (70) maps to a constant $\langle i\lambda\rangle = \mu^2 - \frac{1}{4}$ on $AdS_3$ of radius 1.) The advantage of performing the calculation of the free-energy on $HS^3$ rather than on $AdS_3$ to extract the central charge $a$ is that on $HS^3$ the calculation of the free-energy for the "irregular" symmetric ($S$) modes comes on the same footing as for the "regular" antisymmetric ($A$) modes, while on $AdS_3$ the result for the "irregular" modes was obtained by analytic continuation in $\hat{\Delta} - 1$. [22]

With these remarks in mind, combining the contributions of $S$ and $A$ modes to (71) we find that $F_{S^3,int}$ contains no $\log R$ term, i.e. $a_{int}/N = 0$ for all $\mu$ at $N = \infty$. As already noted, this matches the expectation $a_{int} = 0$ at the special interface fixed point $\mu = 1/2$. The $\mu \to 0$ limit (ordinary interface fixed point) also matches the value $a_{int}^O = 2a_{bound}^O$, where $a_{bound}^O/N$ was found to vanish at $N = \infty$ in Ref. [22]. Finally, we can understand the limit $\mu \to 1$ in the following way. At finite $N$ the extraordinary-log phase ($\mu \to 1$) is described by Eq. (11). Ignoring the coupling term $s$, this corresponds to $N-1$ copies of a free boson $\vec{\pi}$ and two copies of the normal boundary fixed point. Thus, to leading order in the radius $R$, we expect the free energy $F_{S^3,int}$ for the extraordinary-log phase to have the form (65), with

$$a_{int}^{eo} = 2a_{bound}^N + N - 1, \tag{73}$$

where $a_{bound}^N$ is the $a$-coefficient for the normal fixed point in the boundary geometry and the second term comes from the central charge of $N-1$ free 2d bosons. Ref. [22] found that

$$a_{bound}^N = -\frac{N}{2}, \qquad N \to \infty, \tag{74}$$

so Eq. (73) again confirms that $a_{int}^{eo}(N)/N$ is 0 for $N = \infty$. We leave the question of corrections to Eq. (65) in the extraordinary-log phase coming from the logarithmically running coupling $g$ to future work.

## 4.2 Central charge at the ordinary fixed point at $O(N^0)$

We now directly compute the central charge $a_{bound}^O$ for the ordinary boundary fixed point to $O(N^0)$ (i.e. to first subleading order in $N$) by computing the partition function $Z_{HS^3}$. From this, we can obtain the central charge at the ordinary interface fixed point $a_{int}^O = 2a_{bound}^O$. We begin with the action:

$$S = \frac{1}{2} \int_{HS^3} d^D x \sqrt{g} \left[ \partial_\mu \phi_a g^{\mu\nu} \partial_\nu \phi_a + i\lambda \left( \phi_a \phi_a - \frac{1}{g} \right) + \frac{(D-2)}{4(D-1)} \mathcal{R} \phi_a \phi_a \right]. \tag{75}$$

We work around the large-$N$ saddle point $i\lambda = i\lambda_0 + \delta\lambda$,

$$i\lambda_0 = -\frac{1}{4} R^{-2} \sec^2 \alpha. \tag{76}$$

This is the right-hand-side of (70) with $\mu = 0$. To order $N^0$,

$$Z_{HS^3, bound} = \left[ \det\left( -\Delta + i\lambda_0 + \frac{3}{4R^2} \right) \right]^{-N/2} \int D\delta\lambda$$
$$\times \exp\left( -\frac{1}{2} \int d^3 x d^3 y \sqrt{g_x} \sqrt{g_y} \delta\lambda(x) K_\lambda(x,y) \delta\lambda(y) \right). \tag{77}$$

Here $K_\lambda(x,y) = \frac{N}{2} G_0^2(x,y)$ and $G_0(x,y) \delta^{ab} = \langle \phi^a(x) \phi^b(y) \rangle_{HS^3}$ at $N = \infty$. Thus,

$$F_{HS^3, bound}^O = \frac{N}{2} \text{Tr} \log\left( -\Delta + i\lambda_0 + \frac{3}{4R^2} \right) + \frac{1}{2} \text{Tr} \log K_\lambda = -\frac{1}{2} \text{Tr} \log D_\lambda, \tag{78}$$

where we used Eq. (72). Here, the operator $D_\lambda = K_\lambda^{-1}$ is the $\lambda$ propagator to $O(1/N)$,

$$D_\lambda(x,y) = \langle \lambda(x)\lambda(y) \rangle_{HS^3, c} = \Omega^{-2}(x) \Omega^{-2}(y) \langle \lambda(x)\lambda(y) \rangle_{\mathbb{R}_+^3, c}$$
$$= -\frac{1}{\pi^2 N R^4} (x^2 + x_3^2 + 1)^2 (y^2 + y_3^2 + 1)^2 \tag{79}$$
$$\times \left( \frac{1}{((x-y)^2 + (x_3-y_3)^2)^2} - \frac{1}{((x-y)^2 + (x_3+y_3)^2)^2} \right). \tag{80}$$

Thus,

$$D_\lambda(x,y) = D_\lambda^0(x,y) - D_\lambda^0(x, R_3 y), \tag{81}$$

where $D_\lambda^0(x,y)$ is the propagator on the full sphere $S^3$:

$$D_\lambda^0(x,y) = -\frac{16}{\pi^2 N} \frac{1}{s(x,y)^4}, \qquad s(x,y)^2 = 4R^2 \frac{(x-y)^2}{(x^2+1)(y^2+1)}. \tag{82}$$

Here, $s(x,y)$ is the chord distance on the sphere. In Eq. (81), $R_3(y, y_3) = (y, -y_3)$ is the reflection across the equator of $S^3$. Interestingly, the $\lambda$ propagator, Eq. (81), takes a simple Dirichlet-like form to leading order in $N$ – we use this fact shortly.

To compute the trace in (78) we find eigenvalues of $D_\lambda$ on $HS^3$. Due to the Dirichlet-like form of (81), this is equivalent to finding eigenfunctions of $D_\lambda^0$ on the full $S^3$ which are odd under the reflection $R_3$. By rotational symmetry, eigenfunctions of $D_\lambda^0$ are angular harmonics $Y_{n\ell m}$ on $S^3$. Here $-\Delta Y_{n\ell m} = n(n+2)Y_{n\ell m}$, $n = 0, 1, 2, \ldots$, and $\ell = 0, 1, 2, \ldots n$, $m = -\ell, -\ell+1, \ldots \ell$. The eigenvalue of $Y_{n\ell m}$ under the reflection $R_3$ is $(-1)^{n+\ell}$.[10] It was shown in Ref. [28] that

$$\frac{1}{s(x,y)^{2\Delta}} = \frac{1}{R^{2\Delta}} \sum_{n\ell m} g_n Y_{n\ell m}(x) Y^*_{n\ell m}(y), \tag{83}$$

where the eigenvalue

$$g_n = \pi^{D/2} 2^{D-\Delta} \frac{\Gamma(D/2-\Delta)}{\Gamma(\Delta)} \frac{\Gamma(n+\Delta)}{\Gamma(D+n-\Delta)} \rightarrow -4\pi^2(n+1), \tag{84}$$

where we have substituted dimension $D = 3$ and $\Delta = 2$. Thus,

$$F^O_{HS^3,bound} = -\frac{1}{2} \sum_{n=0}^\infty d_n \log E_n, \tag{85}$$

where $E_n = \frac{64}{NR}(n+1)$ and $d_n = \frac{1}{2}n(n+1)$ is the degeneracy of level $n$ eigenstates with $R_3 = -1$. Using $\zeta$-function regularization, we obtain to logarithmic accuracy in $R$

$$F^O_{HS^3,bound} = \frac{1}{2}\frac{d}{ds} \sum_{n=0}^\infty d_n(E_n)^{-s} = \frac{1}{2}\log R \sum_{n=1}^\infty \frac{d_n}{(n+1)^s} = \frac{1}{4}\log R \sum_{n=0}^\infty n(n+1)^{1-s}$$

$$= \frac{1}{4}\log R(\zeta(s-2) - \zeta(s-1)) \rightarrow \frac{1}{48}\log R, \tag{86}$$

where the limit $s \rightarrow 0$ is understood throughout. Thus,

$$a^O_{bound} = -\frac{1}{16} + O(N^{-1}), \tag{87}$$

and

$$a^O_{int} = 2a^O_{bound} = -\frac{1}{8} + O(N^{-1}). \tag{88}$$

## 4.3 Central charge at the normal fixed point at $O(N^0)$

We now directly compute the central charge at the normal boundary fixed point to $O(N^0)$. We follow Refs. [1, 22, 25]. We choose the symmetry breaking field on the boundary to be along the $N$-th direction. We first recall a few facts about the normal fixed point on $\mathbb{R}^3_+$. At $N = \infty$ we have

$$\langle i\lambda(x) \rangle_{\mathbb{R}^3_+} = \frac{3}{4(x^3)^2},$$

$$\langle \phi_N \rangle_{\mathbb{R}^3_+} = \frac{a^0_\sigma}{\sqrt{2x^3}}, \qquad (a^0_\sigma)^2 = \frac{N}{2\pi},$$

$$\langle \lambda(x)\lambda(y) \rangle_{\mathbb{R}^3_+,c} = -\frac{16}{\pi^2 N} \left( \frac{1}{((x-y)^2 + (x^3 - y^3)^2)^2} - \frac{1}{((x-y)^2 + (x^3 + y^3)^2)^2} \right). \tag{89}$$

---

[10]These results can be straightforwardly obtained from the discussion around Eqs. (D.3), (D.4), (D.5) by setting $\mu = 1/2$.

Note that the $\lambda$ propagator at the normal boundary fixed point is the same as at the ordinary boundary fixed point. Making a conformal transformation to $HS^3$, at $N = \infty$

$$
i\lambda_0(x) \equiv \langle i\lambda(x)\rangle_{HS^3} = \Omega(x)^{-2}\langle i\lambda(x)\rangle_{\mathbb{R}^3_+} = \frac{3}{4R^2}\sec^2\alpha,
$$

$$
\sigma_0(x) \equiv \langle\phi_N(x)\rangle_{HS^3} = \Omega(x)^{-1/2}\langle\phi_N\rangle_{\mathbb{R}^3_+} = \frac{a^0_\sigma}{\sqrt{2R}}(\sec\alpha)^{1/2}. \tag{90}
$$

The connected $\lambda$ two-point function on $HS^3$ at $N = \infty$ is given by the same expression as for the ordinary fixed point (80). To compute the partition function on $HS^3$, we expand $\lambda(x) = \lambda_0(x) + \delta\lambda(x)$, $\phi_N(x) = \sigma_0(x) + \delta\sigma(x)$,

$$
S = \int d^d x \sqrt{g}\left[\frac{1}{2}\sum_{i=1}^{N-1}\phi_i\left(-\Delta + i\lambda_0 + \frac{3}{4R^2} + i\delta\lambda\right)\phi_i + \frac{1}{2}\delta\sigma\left(-\Delta + i\lambda_0 + \frac{3}{4R^2} + i\delta\lambda\right)\delta\sigma\right.
$$

$$
\left. + \frac{i\sigma_0^2}{2}\delta\lambda + i\sigma_0\delta\lambda\delta\sigma\right]. \tag{91}
$$

Integrating $\phi_i$ and $\delta\sigma$ out, to first subleading order in $N$ we obtain Eq. (77), where now $K_\lambda(x, y) = \frac{N}{2}G_0^2(x, y) + \sigma_0(x)G_0(x, y)\sigma_0(y)$ and $G_0(x, y) = (-\Delta + i\lambda_0 + \frac{3}{4R^2})^{-1}$ is the two-point function of $\phi_i$, $i = 1, 2\ldots N-1$. Furthermore, $K_\lambda = D_\lambda^{-1}$, with $D_\lambda$ given by Eq. (80). Therefore,

$$
F^N_{HS^3, bound} = \frac{N}{2}\text{Tr}\log\left(-\Delta + i\lambda_0 + \frac{3}{4R^2}\right) - \frac{1}{2}\text{Tr}\log D_\lambda = \frac{1}{6}\left(N + \frac{1}{8}\right)\log R. \tag{92}
$$

Here, we use Eq. (72) with $\mu = 1$ to evaluate the first trace (we use the $A$ branch to recover the correct correlation functions at the normal fixed point) and Eq. (86) to evaluate the second trace. Therefore,

$$
a^N_{bound} = -\frac{N}{2} - \frac{1}{16} + O(N^{-1}). \tag{93}
$$

From this, we obtain the interface central charge at the extraordinary-log fixed point to $O(N^0)$:

$$
a^{eo}_{int} = 2a^N_{bound} + N - 1 = -\frac{9}{8} + O(N^{-1}). \tag{94}
$$

## 4.4 Interface central charge and the $a$-theorem

Finally, we compare the results of the explict calculation of the central charge at the ordinary (88) and extraordinary-log (94) interface fixed points to a detailed form of the $a$-theorem by Jensen and O'Bannon. [17] Through this comparison, we verify our result for the full $\beta$-function (53).

As shown in Ref. [17], for an interface RG flow between a UV and an IR fixed point,

$$
a_{\text{UV}} - a_{\text{IR}} = 3\pi\int d^2x\, x^2\langle\mathcal{T}(x)\mathcal{T}(0)\rangle_c. \tag{95}
$$

Here we are in a configuration with a planar interface at $z = 0$, the integral is over the $z = 0$ plane and the trace of the energy momentum tensor is

$$
T^\mu_\mu = \delta(z)\mathcal{T}. \tag{96}
$$

The correlator in (95) is evaluated in a theory slightly perturbed from the interface UV fixed point. If we write

$$
S = S_{\text{UV}} + g\int d^2x\,\hat{O}(x), \tag{97}
$$

with $\hat{O}(x)$ - the operator perturbing the theory away from the UV interface fixed point, then

$$\mathcal{T} = \beta(g)\hat{O}(x), \tag{98}$$

where $\beta(g)$ is the $\beta$-function for the parameter $g$. We note that Eq. (95) has the same form as the usual Zamolodchikov's $c$-theorem in a purely 2d theory.

To apply the theorem (95) to our set up, consider the flow from the special interface fixed point (UV) to the ordinary fixed point or the extraordinary-log fixed point (IR). We have already explicitly computed the central charges on the left hand side of (95) to $O(N^0)$, see Eq. (9). We now confirm by an explicit calculation that the right hand side of (95) reproduces the same central charge difference.

To do this, we first consider a slightly more general situation. Imagine a theory with a small expansion parameter $\kappa$ (in our case $\kappa = 1/N$). Suppose that at $\kappa = 0$ the theory possesses a line of interface fixed points with the action (97). At $\kappa = 0$ the coupling $g$ parameterizing the line of fixed points is exactly marginal and

$$\langle \hat{O}(\mathrm{x})\hat{O}(0)\rangle_{\mathrm{c}} = \frac{C(g)}{\mathrm{x}^4}, \qquad \kappa = 0. \tag{99}$$

Here $C(g)$ is the Zamolodchikov metric. At small $\kappa$, the coefficient of the $\beta$-function $\beta(g)$ is $O(\kappa)$ and the line of fixed points is lifted so that only several isolated fixed points survive. Let us consider the flow from $g_{\mathrm{UV}}$ to $g_{\mathrm{IR}}$ and use (95) to compute the change of the central charge along this flow. The operator $\hat{O}$ in (97) acquires an anomalous dimension along the flow. Under an RG scale transformation by $d\ell$, $\hat{O}(\mathrm{x}) \to (1 - \beta'(g)d\ell)\hat{O}((1 - d\ell)\mathrm{x})$. Thus, the two point function $G_{\hat{O}}(\mathrm{x}) = \langle \hat{O}(\mathrm{x})\hat{O}(0)\rangle_c$ satisfies the Callan-Symanzik equation:

$$\left(\Lambda\frac{d}{d\Lambda} + \beta(g)\frac{d}{dg} + 2\gamma_{\hat{O}}(g)\right)G_{\hat{O}}(g,\Lambda,x) = 0, \quad \gamma_{\hat{O}}(g) = \beta'(g), \tag{100}$$

where $\Lambda$ is the UV cut-off. Solving the Callan-Symanzik equation, to leading order in $\kappa$,

$$G_{\hat{O}}(g_0,\Lambda,\mathrm{x}) \approx \frac{1}{\mathrm{x}^4}Z^2(\ell)C(g(\ell)), \qquad \ell = \log \Lambda\mathrm{x}, \tag{101}$$

where $C(g)$ is given by Eq. (99) and

$$\frac{dg}{d\ell} = -\beta(g(\ell)), \qquad g(0) = g_0,$$

$$\log Z(\ell) = -\int_0^\ell d\ell' \beta'(g(\ell')) = \int_{g_0}^{g(\ell)} dg' \frac{\beta'(g')}{\beta(g')} = \log\frac{\beta(g(\ell))}{\beta(g_0)}. \tag{102}$$

Evaluating Eq. (95) in a theory slightly perturbed from the interface UV fixed point,

$$a_{\mathrm{UV}} - a_{\mathrm{IR}} \approx 3\pi \int_{\Lambda|x|>1} \frac{d^2\mathrm{x}}{\mathrm{x}^2} \beta(g_{\mathrm{UV}})^2 Z^2(\log \Lambda x)C(g(\log \Lambda x)) \tag{103}$$

$$= 3\pi \int_{\Lambda|x|>1} \frac{d^2\mathrm{x}}{\mathrm{x}^2} \beta(g(\log \Lambda x))^2 C(g(\log \Lambda x))$$

$$= 6\pi^2 \int_0^\infty d\ell\, \beta(g(\ell))^2 C(g(\ell)) = -6\pi^2 \int_{g_{\mathrm{UV}}}^{g_{\mathrm{IR}}} dg' \beta(g')C(g'). \tag{104}$$

Let's consider re-parameterizing $g \to g(u)$, with $u$ – a new coupling constant. If we make an infinitesimal change, $u \to u + \delta u$,

$$\delta S = \delta u \frac{dg}{du} \int d^2\mathrm{x}\,\hat{O}(\mathrm{x}). \tag{105}$$

Thus, the operator conjugate to $u$ is $\frac{dg}{du}\hat{O}(x)$, which has the Zamolodchikov norm $C_u = \left(\frac{dg}{du}\right)^2 C(g(u))$. Likewise, $\beta_u = (\frac{dg}{du})^{-1}\beta(g(u))$. Thus, Eq. (104) is invariant under reparametrization:

$$a_{\text{UV}} - a_{\text{IR}} \approx -6\pi^2 \int_{u_{\text{UV}}}^{u_{\text{IR}}} du \, \beta_u(u) C_u(u). \qquad (106)$$

It can be checked that Eq. (104) reproduces the correct $a_{\text{UV}} - a_{\text{IR}}$ for the case of short RG flows analyzed in Ref. [22], where $C(g)$ is, to leading order in $\kappa$, constant along the flow.

Let us now apply (104) to our problem of the interface in the $O(N)$ model. At $N = \infty$ we have the coupling $\mu$ parametrizing the line of fixed points. We already know the $\beta$-function, $\beta(\mu)$, Eq. (53), to $O(1/N)$. It remains to compute the Zamolodchikov norm of the operator conjugate to $\mu$. The marginal operator that tunes the system along the line of fixed points is just $i\lambda_S(x^3 = 0)$ (recall its bulk to boundary OPE contains an operator of dimension $\hat{\Delta} = 2$, see section 3.1). Upon varying $\mu$, we have

$$\delta S = \delta\mu \cdot \xi(\mu) \int d^2x \, i\lambda_S \left(x, x^3 = 0\right), \qquad (107)$$

where $\xi(\mu)$ is a to be determined function. From (25), we know the response of $\langle i\lambda_S(x^3)\rangle$ to variations in $\mu$:

$$\delta\langle i\lambda_S(x^3)\rangle = \frac{2\sqrt{2}\mu\delta\mu}{(x^3)^2}. \qquad (108)$$

Performing perturbation theory in $\delta\mu$,

$$\delta\langle i\lambda_S(x^3)\rangle = -\delta\mu \cdot \xi(\mu) \int d^2x \, \langle i\lambda_S(0, x^3) i\lambda_S(x, 0)\rangle_{\text{c}}. \qquad (109)$$

Using (43), we get

$$\xi(\mu) = -\frac{\sqrt{2}\pi N\mu}{16\sin^2 \pi\mu}, \qquad C(\mu) = \frac{32\sin^2 \pi\mu}{\pi^2 N}\xi^2(\mu) = \frac{N}{4}\frac{\mu^2}{\sin^2 \pi\mu}. \qquad (110)$$

Now, substituting $C(\mu)$ above and $\beta(\mu)$ into Eq. (104), we obtain

$$a_{int}^{sp} - a_{int}^{O} \approx -6\pi^2 \int_{1/2}^{0} d\mu \, \beta(\mu)C(\mu) = \frac{1}{8} + O(N^{-1}),$$

$$a_{int}^{sp} - a_{int}^{eo} \approx -6\pi^2 \int_{1/2}^{1} d\mu \, \beta(\mu)C(\mu) = \frac{9}{8} + O(N^{-1}). \qquad (111)$$

As previously discussed, $a_{int}^{sp} = 0$. Thus, we recover the results (88) and (94) obtained by an explicit calculation.

## 5 $\beta$-function in the semi-infinite geometry

In this section, we return to the problem of the $O(N)$ model in a semi-infinite geometry. As we mentioned in the introduction, two scenarios for the evolution of the phase diagram past $N = N_c$ were proposed in Ref. [1], see Fig. 5. Which scenario is realized is determined by the sign of a higher order term in the $\beta$-function for the surface spin-stiffness. In this section, we determine this higher order term in the limit $N \to \infty$. Instead of computing the $\beta$-function directly, we extract the higher order term by matching the RG treatment of Ref. [1] to known

large-$N$ results on the special boundary fixed point in bulk dimension $D > 3$. [14] Thus, we consider a $d$ dimensional boundary of a $d+1$ dimensional bulk. We begin with the action

$$S = S_{\text{norm}} + S_{\vec{n}} - s \int d^d x \, \pi_i \mathfrak{t}_i \,, \tag{112}$$

with

$$S_{\vec{n}} = \int d^d x \left[ \frac{1}{2g} \left( (\partial_\mu \vec{\pi})^2 + \frac{1}{1-\vec{\pi}^2} (\vec{\pi} \cdot \partial_\mu \vec{\pi})^2 \right) - \vec{h} \cdot \vec{n} \right]. \tag{113}$$

Here, we've added a symmetry breaking field $\vec{h} = h\hat{e}_N$ as an infra-red regulator. Here and below, $d = 2+\epsilon$ denotes the surface dimension, while $D = d+1$ stands for the bulk dimension. We are interested in the limit $\epsilon \ll 1$. An argument analogous to that in Ref. [1] gives

$$s^2 = \frac{\Gamma(d)^2}{(4\pi)^d \Gamma(d/2)^2} \frac{a_\sigma^2}{b_{\mathfrak{t}}^2} \,, \tag{114}$$

in terms of the OPE coefficients $a_\sigma$, $b_{\mathfrak{t}}$ of the normal boundary universality class:

$$\phi_N(\mathrm{x}, x^3) \sim \frac{a_\sigma}{(2x^3)^{\Delta_\phi}} + b_\mathrm{D}(2x^3)^{d+1-\Delta_\phi} \mathrm{D}(\mathrm{x}) + \dots, \qquad x^3 \to 0, \tag{115}$$

$$\phi_i(\mathrm{x}, x^3) \sim b_{\mathfrak{t}}(2x^3)^{d-\Delta_\phi} \mathfrak{t}_i(\mathrm{x}) + \dots, \qquad x^3 \to 0. \tag{116}$$

Note that $s^2$ depends on $d$.

As discussed in Ref. [1], the leading terms in $\beta(g)$ are:

$$\beta(g) \approx \epsilon g + \alpha_{\text{bound}} g^2 + b g^3, \qquad \alpha_{\text{bound}} = \frac{\pi s^2(d=2)}{2} - \frac{N-2}{2\pi}. \tag{117}$$

$\alpha_{\text{bound}}(N)$ changes sign at $N = N_c > 2$ from positive at $N < N_c$ to negative at $N > N_c$. For $D = 3$ and $N < N_c$, $g$ flows logarithmically to zero and the extraordinary-log fixed point is realized. The evolution of the phase diagram in $D = 3$ past $N = N_c$ depends on the sign of the coefficient $b$ in (117).

1. If $b < 0$, then for $N \to N_c^-$, we have a perturbatively accessible IR unstable fixed point at $g_* \approx \frac{\alpha_{\text{bound}}}{|b|}$. It is natural to identify this fixed point with the special transition between the extraordinary-log and ordinary phases. At $N = N_c$ the special fixed point annihilates with the extraordinary fixed point at $g = 0$ and only the ordinary fixed point remains for $N \geq N_c$, see Fig. 11 (left).

2. If $b > 0$, then for $N \to N_c^+$, the extraordinary fixed point moves away from $g = 0$ to $g_* \approx \frac{|\alpha_{\text{bound}}|}{b}$. Thus, we find an IR stable conformal fixed point for $N$ just above $N_c$, which we term the extraordinary-power fixed point. Since only the ordinary fixed point is found by large-$N$ calculations in $D = 3$, the extraordinary-power fixed point presumably annihilates with the special fixed point at some larger value of $N = N_{c2}$, see Fig. 11 (right).

From the form of the action (112), a direct computation of the coefficient $b$ in $\beta(g)$ requires the knowledge of the four-point function of the tilt operator $\mathfrak{t}_i$ at the normal fixed point. (This should be compared to the computation of the coefficient $\alpha_{\text{bound}}$, which relies only on the two-point function of $\mathfrak{t}_i$ and the knowledge of the coefficient $s$.) In addition, a number of higher order counter-terms in the action, omitted in Eq. (112), such as e.g. $\delta L_{\text{bound}} \sim \vec{\pi}^2 \pi_i \mathfrak{t}_i$, would have to be fixed by the requirement of O($N$) invariance. We do not pursue this route to computing $b$ here.

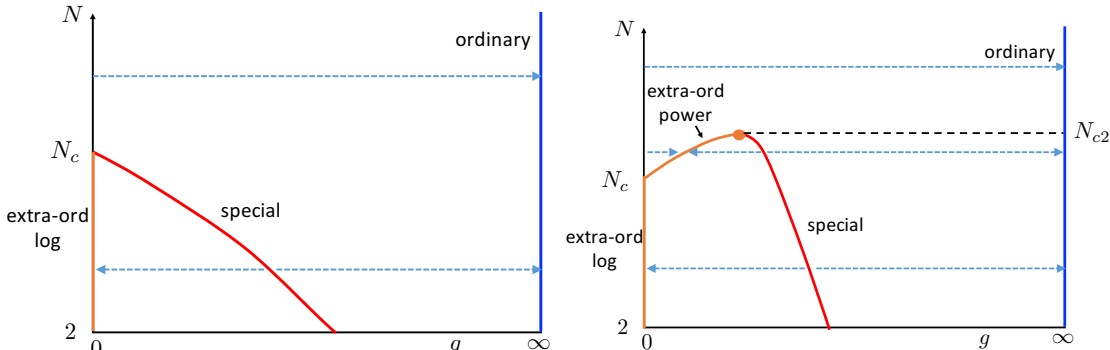

Figure 11: Conjectured RG flows of the semi-infinite O($N$) model in $D = 3$. Left - scenario I. Right - scenario II. Blue dashed arrows indicate the direction of RG flow. Black dashed lines are guide to eye.

Instead, we compute $b(N)$ for $N \to \infty$ by considering the special transition in $D = 3 + \epsilon$. Here, the $g = 0$ fixed point is always stable — it describes an extraordinary phase with true long range boundary order. For $N \gtrsim N_c$, (117) gives an IR unstable fixed point at

$$g_*^{\text{spec}} \approx \frac{\epsilon}{|\alpha_{\text{bound}}|} + b \frac{\epsilon^2}{|\alpha_{\text{bound}}|^3} + O(\epsilon^3). \tag{118}$$

We identify this fixed point with the special transition separating the extraordinary and the ordinary phases. The scaling dimension of the boundary order parameter at this fixed point is given by

$$\Delta_{\vec{n}}^{\text{spec}} = \frac{\eta_{\vec{n}}(g_*^{\text{spec}})}{2}, \tag{119}$$

with

$$\eta_{\vec{n}}(g) = \frac{(N-1)g}{2\pi} + O(g^2) \tag{120}$$

the anomalous dimension of $\vec{n}$. At the same time, the special fixed point is accessible with the large-$N$ expansion for any dimension $D$ in the range $3 < D < 4$, in particular, the scaling dimension of the surface order parameter $\Delta_{\vec{n}}^{\text{spec}}$ has been computed to $O(1/N)$ [14],

$$\begin{aligned}
\Delta_{\vec{n}}^{\text{spec}} &= D - 3 + \frac{1}{N} \frac{2(4-D)}{\Gamma(D-3)} \left[ \frac{(6-D)\Gamma(2D-6)}{D\Gamma(D-3)} + \frac{1}{\Gamma(5-D)} \right] + O\left(\frac{1}{N^2}\right) \\
&= \epsilon + \frac{1}{N}(3\epsilon - \frac{5}{3}\epsilon^2 + O(\epsilon^3)) + O(N^{-2}). \tag{121}
\end{aligned}$$

Thus, for $\epsilon \to 0$ and $N \to \infty$, we can compare the predictions of our RG analysis to the direct large-$N$ expansion. To leading order in $\epsilon$, this was already done in Ref. [1]: $\Delta_{\vec{n}}^{\text{spec}}$ found from (15), (118), (120) matches exactly with (121) to $O(\epsilon)$, including the subleading $O(1/N)$ term. We aim to match (121) with the RG analysis to $O(\epsilon^2)$ and $O(1/N)$. More specifically, we compute the anomalous dimension $\eta_{\vec{n}}(g)$ to order $g^2$. This can be computed without any extra data for the normal transition, besides the coefficient $s^2(d)$. Then, we substitute our expression for $g_*^{\text{spec}}$ from (118) into (119) and compute $b$ in the limit $N \to \infty$ by matching to (121).

We now compute $\eta_{\vec{n}}$ to order $g^2$. The coefficient of the $g^2$ term in $\eta_{\vec{n}}$ is scheme dependent. We use dimensional regularization with the following conventions:

$$\tilde{g} = \mu^{-\epsilon} Z_g(\tilde{g}_r)\tilde{g}_r, \qquad \vec{n} = Z_{\vec{n}}^{1/2}\vec{n}_r, \qquad \vec{h} = Z_{\vec{n}}^{-1/2}\vec{h}_r, \tag{122}$$

with

$$Z_g = 1 + \sum_{m=1}^{\infty} \sum_{k=1}^{m} \frac{Z_g^{m,k}}{\epsilon^k} \tilde{g}_r^m, \qquad Z_{\vec{n}} = 1 + \sum_{m=1}^{\infty} \sum_{k=1}^{m} \frac{Z_{\vec{n}}^{m,k}}{\epsilon^k} \tilde{g}_r^m, \tag{123}$$

and

$$\tilde{g} = \frac{2}{(4\pi)^{d/2}\Gamma(d/2)} g. \tag{124}$$

The $\beta$-function, $\beta(\tilde{g})$ and anomalous dimension $\eta_{\vec{n}}$ are obtained from renormalization constants $Z_g$, $Z_{\vec{n}}$ using

$$\beta(\tilde{g}_r) = \mu \frac{\partial}{\partial \mu} \tilde{g}_r \Big|_{g,\Lambda} = \epsilon \left[ \frac{d}{d\tilde{g}_r} \log\left(\tilde{g}_r Z_g(\tilde{g}_r)\right) \right]^{-1}, \tag{125}$$

$$\eta_{\vec{n}}(\tilde{g}_r) = \mu \frac{\partial}{\partial \mu} \log Z_{\vec{n}} \Big|_{g,\Lambda} = \beta(\tilde{g}_r) \frac{d}{d\tilde{g}_r} \log Z_{\vec{n}}(\tilde{g}_r). \tag{126}$$

The renormalized correlation function of $m$ $\vec{n}$ fields, $D_r^m = Z_{\vec{n}}^{-m/2} D^m$, then satisfies,

$$\left( \mu \frac{\partial}{\partial \mu} + \beta(\tilde{g}_r) \frac{\partial}{\partial \tilde{g}_r} + \frac{m}{2} \eta_{\vec{n}}(\tilde{g}_r) \right) D_r^m(\tilde{g}_r, \mu) = 0. \tag{127}$$

The constants $Z_g$, $Z_{\vec{n}}$ can be found by computing the two-point function of $\vec{\pi}$ and the one-point function of $n_N$. Let $\langle \pi^i(\mathrm{x})\pi^j(0) \rangle = \delta^{ij} D(x)$. Then to order $g^2$,

$$D(p) = D_0(p) + \delta_{\mathrm{NL\Sigma M}} D(p) + \delta_s D(p) + O(g^3). \tag{128}$$

$D_0(p) = \frac{g}{p^2+m^2}$, with $m^2 = gh$, is the free propagator. $\delta_{\mathrm{NL\Sigma M}}D(p)$ is the standard contribution from the leading non-linear terms in $S_{\vec{n}}$, while $\delta_s D(p)$ is the leading contribution from the coupling $s$ to the operators of the normal fixed point. Evaluating these, we obtain

$$\delta_{\mathrm{NL\Sigma M}}D(p) = -\frac{\Gamma(1-d/2)}{(4\pi)^{d/2}} \frac{g^2 m^{d-2}}{p^2+m^2} \left( 1 + \frac{(N-3)m^2}{2(p^2+m^2)} \right), \tag{129}$$

$$\delta_s D(p) = \frac{s^2 \Gamma(-d/2) \pi^{d/2}}{2^d \Gamma(d)} \frac{g^2 p^{d-2}}{p^2+m^2} \left( 1 - \frac{m^2}{p^2+m^2} \right). \tag{130}$$

Extracting $Z_g$,

$$Z_g = 1 - \tilde{\alpha} \frac{\tilde{g}_r}{\epsilon} + O(\tilde{g}_r^2), \qquad \tilde{\alpha} = \pi^2 s^2(d=2) - (N-2), \tag{131}$$

$$\beta(\tilde{g}_r) = \epsilon \tilde{g}_r + \tilde{\alpha} \tilde{g}_r^2 + \tilde{b} \tilde{g}_r^3 + \dots \tag{132}$$

Note that our normalization for the coefficient $\tilde{\alpha}$ here differs by a factor of $2\pi$ from that of $\alpha$ in (117). Our goal is to compute $\tilde{b}$ in the large-$N$ limit. The value of $\tilde{\alpha}$ starts positive at $N = 2$ and eventually becomes negative for $N > N_c > 2$. [1,7] In particular, in the large-$N$ limit [1]:

$$\pi^2 s^2(d=2) = \frac{N}{2} + O(N^{-1}), \quad \tilde{\alpha} = -\frac{N-4}{2} + O(N^{-1}). \tag{133}$$

When $\tilde{\alpha} < 0$ (i.e. for $N > N_c$) and $\epsilon > 0$ is small, the system has an IR-unstable fixed point at

$$\tilde{g}_r^* = \frac{\epsilon}{|\tilde{\alpha}|} + \tilde{b} \frac{\epsilon^2}{|\tilde{\alpha}|^3} + O(\epsilon^3). \tag{134}$$

We identify this fixed point with the special transition in $d = 2 + \epsilon$.

We next proceed to compute $\eta_{\vec{n}}(\tilde{g}_r)$. We compute the one-point function of $n_N \approx 1 - \frac{1}{2}\vec{\pi}^2 - \frac{1}{8}(\vec{\pi}^2)^2$,

$$\langle n_N \rangle = 1 - \frac{N-1}{2}D(\mathrm{x}=0) - \frac{N^2-1}{8}D_0(\mathrm{x}=0)^2 + O(g^3). \tag{135}$$

Fourier transforming $D(p)$ back to real space,

$$D_0(\mathrm{x}=0) = \frac{g\Gamma(1-d/2)}{(4\pi)^{d/2}}m^{d-2}, \tag{136}$$

$$\delta_{\mathrm{NL\Sigma M}}D(\mathrm{x}=0) = -\frac{g^2\Gamma(1-d/2)^2 m^{2d-4}}{(4\pi)^d}\left(1 + \frac{N-3}{2}(1-d/2)\right), \tag{137}$$

$$\delta_s D(\mathrm{x}=0) = -\frac{g^2 s^2 \pi \csc(\pi(d-2))\Gamma(-d/2)m^{2d-4}}{2^{2d}\Gamma(d-1)\Gamma(d/2)}. \tag{138}$$

Expressing $\langle n_N \rangle$ in terms of $\tilde{g}_r$ and $h_r$, we obtain

$$Z_{\vec{n}} = 1 + \frac{Z_{\vec{n}}^{1,1}}{\epsilon}\tilde{g}_r + \left(\frac{Z_{\vec{n}}^{2,2}}{\epsilon^2} + \frac{Z_{\vec{n}}^{2,1}}{\epsilon}\right)\tilde{g}_r^2 + O(\tilde{g}_r^3), \tag{139}$$

with

$$Z_{\vec{n}}^{1,1} = N-1, \qquad Z_{\vec{n}}^{2,2} = (N-1)\left(N - \frac{3}{2} - \frac{\pi^2 s^2(d=2)}{2}\right), \tag{140}$$

$$Z_{\vec{n}}^{2,1} = \frac{(N-1)\pi^2}{2}\left[\frac{d}{d\epsilon}\left(\frac{2\pi^{d-2}s^2(d)}{d\Gamma(d-1)}\right)\bigg|_{\epsilon=0} - s^2(d=2)\right], \tag{141}$$

so the anomalous dimension

$$\eta_{\vec{n}} = (N-1)\tilde{g}_r + 2Z_{\vec{n}}^{2,1}\tilde{g}_r^2. \tag{142}$$

Thus, to obtain $Z_{\vec{n}}^{2,1}$, we need to know the value of $s^2(d)$ and its derivative at $d=2$. For a general value of $N$, we don't know $s^2$. However, in the large $N$ limit, using the results of Ref. [8],

$$\frac{2\pi^{d-2}s^2(d)}{d\Gamma(d-1)} = \frac{N(d-1)}{2\pi^2\cos(\pi\epsilon/2)}\left(1 + \frac{f(d)}{N} + O(N^{-2})\right), \tag{143}$$

where we have introduced a yet unknown next correction in $N$, parametrized by the function $f(d)$. Then

$$Z_{\vec{n}}^{2,1} = \frac{N}{4}f'(d=2) + O(N^0). \tag{144}$$

Thus, to determine the leading order in $N$ contribution to $Z_{\vec{n}}^{2,1}$, we need to know $f'(2)$. In Ref. [1], we analytically found $f(2) = 0$. In appendix E, we follow the procedure of Refs. [1,25] to compute $f(d)$ in $1 < d < 3$. We were not able to obtain an analytic expression for $f(d)$ and had to resort to numerical integration. We then fitted $f(d)$ near $d=2$ to find

$$f'(2) = \frac{11}{3} \pm 10^{-2}, \tag{145}$$

where the estimated uncertainty is due to numerical integration. We don't currently know if $f'(2) = \frac{11}{3}$ exactly. Thus, we obtain

$$\eta_{\vec{n}}(\tilde{g}_r) = (N-1)\tilde{g}_r + \left(\frac{f'(2)}{2}N + O(N^0)\right)\tilde{g}_r^2 + O(\tilde{g}_r^3). \tag{146}$$

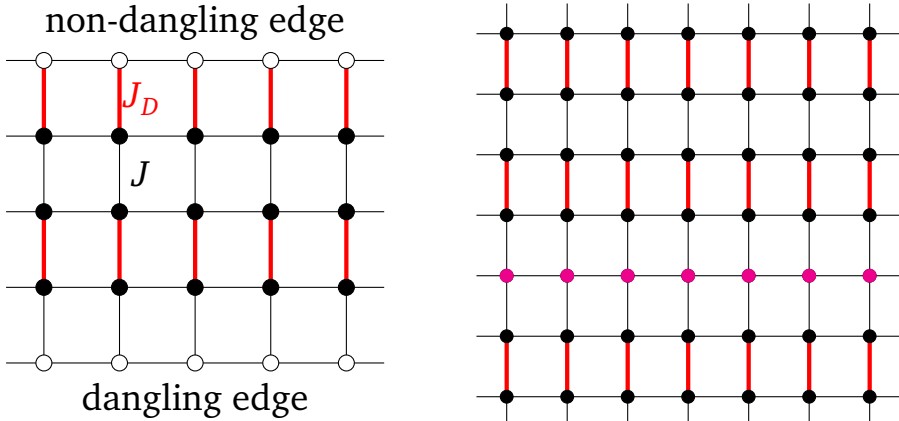

Figure 12: Left: The quantum spin model in Eq. (149) terminated above and below. The red couplings have strength $J_D$ and the black couplings have the strength $J$. The top edge is termed "non-dangling" while the bottom edge is termed "dangling". Right: The quantum spin model (149) with no edges and an inserted row of spins in the third to last shown row.

From the Callan-Symanzik equation, the dimension of $\vec{n}$ at the special transition in $d = 2 + \epsilon$ is given by Eq. (119) with $\tilde{g}_r^*$ given by Eq. (134).

Now, we can match our renormalization group result to that obtained using direct large-$N$ expansion for the special transition, Eq. (121). Matching this to Eqs. (119), (146), (134), we obtain

$$\tilde{b} = \left(-\frac{5}{12} - \frac{f'(2)}{4}\right)N + O(N^0) = \left(-\frac{4}{3} \pm 2.5 \cdot 10^{-3}\right)N + O(N^0). \tag{147}$$

Thus, $\tilde{b}$ is negative for large $N$. Assuming that $\tilde{b}$ remains negative down to $N = N_c$, scenario I discussed in section 1 for the evolution of the phase diagram in $d = 3$ as a function of $N$ is favored. Note that for the nonlinear $\sigma$-model, i.e. $S_{\vec{n}}$ alone (without the coupling to $S_{\text{norm}}$ through the tilt operator), $\tilde{b} = -(N-2)$. [29,30] Thus, the coupling to the bulk only makes $\tilde{b}$ more negative for large $N$. Note that for the plane defect geometry, from Eq. (55),

$$\beta(\tilde{g}) \approx 2\tilde{g}^2 - \frac{5}{3}N\tilde{g}^3, \tag{148}$$

i.e. for $N \to \infty$, $\tilde{b}$ is shifted by $-N/3$ in the semi-infinite geometry compared to the pure 2d model, and by $-2N/3$ in the plane defect geometry.

## 6 Future directions: quantum models

In this paper we have focused on boundary and interface behavior in the classical $O(N)$ model. What happens in the quantum generalization of this problem, i.e, quantum spin systems in two spatial dimensions that undergo an $O(3)$ transition in the bulk? A prototypical Hamiltonian exhibiting such a transition is given by a spin $S$ Heisenberg model on a rectangular lattice

$$H = \sum_{\langle ij \rangle} J_{ij} \vec{S}_i \cdot \vec{S}_j, \tag{149}$$

with the nearest neighbour couplings $J_{ij}$ dimerized as in Fig. 12 (left). As one increases the strength of the red bonds relative to the black bonds, the system goes from a Néel antiferromagnet to a trivial paramagnet. The transition between these phases lies in the classical 3D $O(3)$ universality class as confirmed by numerical calculations. [31]

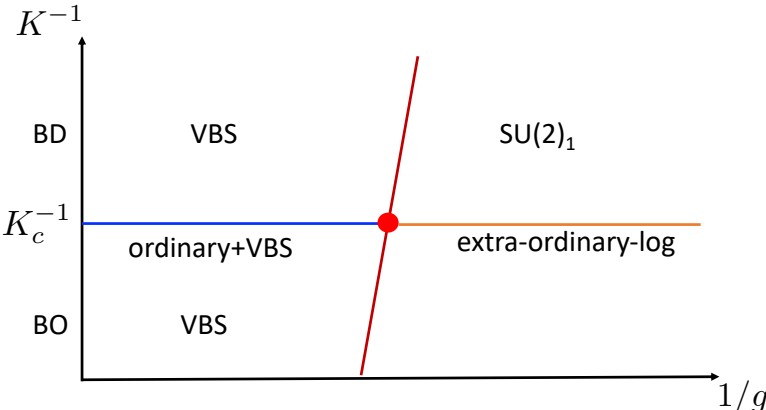

Figure 13: Candidate phase diagram for both the dangling edge and inserted row of spins of a half-integer spin quantum dimer model. The vertical axis corresponds to the bulk coupling and the horizontal axis to the defect coupling.

Boundary behavior in the model (149) (and in similar models) at the bulk critical point has been studied both numerically [32–40] and analytically [1,41]. This model has two possible kinds of edges, "dangling" and "non-dangling", see Fig. 12 (left). When the spin $S$ is an integer, one theoretically expects the universal properties of both the dangling and non-dangling edges to coincide with those of the classical 3D $O(3)$ model. However, when the spin $S$ is a half-integer, one expects only the non-dangling edge to be described by classical $O(3)$ boundary universality. On the paramagnetic side of the phase diagram, the dangling edge is described by a 1d spin-$S$ chain and should either be gapless or break the translational symmetry along the edge by the Lieb-Schultz-Mattis theorem. Such an edge feature is clearly absent in the classical $O(3)$ model. One theoretical possibility for the phase diagram of the dangling edge for half-integer $S$ is shown in Fig. 13. Here the extraordinary-log phase has the same universal properties as for the classical $O(3)$ model, and the ordinary+VBS phase corresponds to the ordinary universality class of the classical $O(3)$ model coexisting with valence bond solid (VBS) boundary order. The "special" transition between these boundary phases is, in principle, different from the special transition in the classical $O(3)$ model, although the critical exponents for the two can be numerically close.[11] We note, however, that current numerical simulations of the model (149) and of similar models do not fully agree with the above theoretical picture for either the dangling or the non-dangling edge. We do not attempt to reconcile the analytical picture above with numerics.

Instead, we briefly comment on a 1d interface defect in the quantum model (149). One way to obtain such a defect is to change the couplings $J_{ij}$ for several rows of spins. This should correspond to perturbing the $O(3)$ model by a local operator along a 2d space-time slice, so we expect the same phase diagram and universal properties as for an interface in the classical 3D $O(3)$ model. A different type of defect arises when one inserts a row of spins along the interface, see Fig. 12 (right). This is the interface analogue of a dangling edge. If the inserted spins are half-integer, the interface again is gapless or breaks translational symmetry even when the bulk is in the paramagnetic phase. Thus, the interface universality must again be distinct from that in a classical model. A possible phase diagram is again given by Fig. 13. The extraordinary-log phase is described by Eq. (11) where the NLΣM action $S_n$ is supplemented

---

[11]Strictly speaking, it is not known whether a continuous special transition exists.

by a topological $\theta$-term:

$$S_\theta = \frac{i\theta}{4\pi} \int dx\, d\tau\, \vec{n} \cdot (\partial_x \vec{n} \times \partial_\tau \vec{n}), \qquad \theta = \pi. \tag{150}$$

Since the $\theta$ term does not affect the perturbative expansion in coupling $g$ of (10), and $g$ runs logarithmically to zero in the extraordinary-log phase, we expect the universal features at the bulk critical point to remain the same as for the interface in the classical model. Likewise, the ordinary+VBS phase is essentially the same as for the ordinary interface in the classical model, apart from an overall two-fold degeneracy of the ground state. However, the transition between the ordinary+VBS and the extraordinary-log interface phases must be different from the special interface fixed point in the classical model. Indeed, let's begin with a decoupled spin-1/2 Heisenberg chain described by a $SU(2)_1$ Wess-Zumino-Witten (WZW) model. Inserting this spin-chain into the bulk of an $O(3)$ model, we find a relevant coupling

$$\delta S = u \int dx\, d\tau\, N^a(x,\tau)\phi^a(x,\tau), \tag{151}$$

where $N^a$ is the Neel order parameter on the Heisenberg chain and $\phi^a$ is the bulk order parameter. Using $\Delta_N = 1/2$, $\Delta_\phi = 0.518936(67)$ [42], we see that the coupling $u$ is relevant: the spin-1/2 chain does not decouple from the bulk.

We may also proceed analogously to Ref. [41]: start with the ordinary interface fixed point of the $O(3)$ model, corresponding to the two ordinary boundary fixed points for the two sides of the 1d chain interface in Fig. 12 (right), and couple in the Heisenberg chain. We obtain:

$$S = S_{\text{ord}}^1 + S_{\text{ord}}^2 + S_{\text{WZW}} + \int dx\, d\tau \left( \lambda J^a \bar{J}^a + uN^a(\hat{\phi}_a^1 + \hat{\phi}_a^2) + v\hat{\phi}_a^1 \hat{\phi}_a^2 \right), \tag{152}$$

with $\hat{\phi}_a^{1,2}$ – the boundary order parameters of the ordinary fixed points and $J^a$, $\bar{J}^a$ – the left/right $SU(2)$ currents of the WZW model. For simplicity, we have assumed reflection symmetry across the inserted chain. The coupling $\lambda$ is marginal at tree level. Using $\Delta_{\hat{\phi}} = 1.187(2)$, [3] the coupling $u$ is slightly relevant: $\dim[u] = 3/2 - \Delta_{\hat{\phi}} = 0.313(2)$. The coupling $v$ is slightly irrelevant: $\dim[v] = -2(\Delta_{\hat{\phi}} - 1) \approx -0.374(4)$. We attempt to directly perform conformal perturbation theory in $u$, $v$. Since $\dim[u]$ and $\dim[v]$ are not infinitesimal, the results are somewhat scheme dependent. We use the scheme in Ref. [43]. We have the following OPE's:

$$J^a(z)\bar{J}^a(\bar{z})J^b(w)\bar{J}^b(\bar{w}) = \frac{3}{4|z-w|^4} - \frac{2}{|z-w|^2}J^c(w)\bar{J}^c(\bar{w}) + \dots,$$

$$N^a(z,\bar{z})N^a(w,\bar{w}) = \frac{3}{2|z-w|}\left(1 + \frac{1}{3}|z-w|^2 J^b(w)\bar{J}^b(\bar{w}) + \dots\right),$$

$$J^a(z)\bar{J}^a(\bar{z})N^b(w,\bar{w}) = \frac{1}{4|z-w|^2}N^b(w,\bar{w}) + \dots,$$

$$\hat{\phi}_a^1(z,\bar{z})\hat{\phi}_b^1(w,\bar{w}) = \frac{\delta^{ab}}{|z-w|^{2\Delta_{\hat{\phi}}}} + \dots, \quad \hat{\phi}_a^2(z,\bar{z})\hat{\phi}_b^2(w,\bar{w}) = \frac{\delta^{ab}}{|z-w|^{2\Delta_{\hat{\phi}}}} + \dots,$$

$$\hat{\phi}_a^1(z,\bar{z})\hat{\phi}_b^2(w,\bar{w}) = \hat{\phi}_a^1 \hat{\phi}_b^2(w,\bar{w}) + \dots \tag{153}$$

Here we've set the velocity of the 1d chain equal to the bulk velocity and only included terms in the OPE with zero Lorentz spin. We obtain the following RG equation for the couplings $\lambda, u, v$:

$$\frac{d\lambda}{d\ell} \approx \pi \left(2\lambda^2 - u^2\right), \tag{154}$$

$$\frac{du}{d\ell} \approx \left(\frac{3}{2} - \Delta_{\hat{\phi}}\right)u - \pi\left(\frac{\lambda}{2} + 2v\right)u, \tag{155}$$

$$\frac{dv}{d\ell} \approx 2\left(1 - \Delta_{\hat{\phi}}\right)v - \pi u^2. \tag{156}$$

Compared to Ref. [41], the coefficient of $u^2$ in (154) is doubled; in addition there is an extra contribution from the coupling $v$ to the flow of $u$, and a flow equation for $v$. Inserting the value of $\hat{\Delta}_{\phi}$, we find no real fixed points with $u \neq 0$. Thus, the present RG approach fails to describe the extaordinary-log to ordinary+VBS interface transition in Fig. 13. We cannot rule out that this transition is first order.

We conclude by noting that it would be interesting to study both 3d classical and 2d quantum spin models with interfaces using Monte Carlo simulations.

**Note Added**: We would like to point out several papers that relate to this work that have appeared after the arXiv publication of this manuscript.

- Refs. [44–46] study a related problem of a two-dimensional surface defect in the $O(N)$ model in general dimension $d$.

- Ref. [47] shows that under reasonable assumptions on the defect RG flow, spontaneous breaking of continuous symmetry (true long range order) on a two-dimensional defect in a $d$-dimensional CFT is not allowed. Instead, depending on the CFT and the symmetry, one may find a non-linear $\sigma$-model on the surface whose "radius" logarithmically flows to infinity under RG, giving rise to surface order parameter correlation functions falling off logarithmically. This is a generalization of the extraordinary-log universality class beyond the $O(N)$ CFT.

- Ref. [48] studies an interface in the classical 3d O(2) model through Monte Carlo simulations. The findings of Ref. [48] are consistent with the interface realizing the extraordinary-log universality class in the regime of large interface coupling. In particular, the value of the $\beta$-function coefficient $\alpha$ extracted numerically from the interface stiffness is found to be $\alpha_{\text{plane}}(N = 2) = 0.56(3)$ and the value of the exponent $q$ of the interface two-point function, Eq. (2), is found to be $q_{\text{plane}}(N = 2) = 0.29(2)$. This agrees with our prediction in table 1, $\alpha_{\text{plane}}(N = 2) = 0.600(10)$ and with the corresponding value of $q_{\text{plane}} = 0.265(4)$, see Eq. (20).

# Acknowledgements

We are grateful to Himanshu Khanchandani, Yifan Wang and Cenke Xu for discussions. We also thank Ilya Gruzberg, Marco Meineri and Jay Padayasi for collaboration on a related project. We thank Gabriel Cuomo, Simone Giombi, Chris Herzog, Kristan Jensen, Chao-Ming Jian, Himanshu Khanchandani, Zohar Komargodski, Marco Meineri, Andy O'Bannon, Michael Smolkin, Yifan Wang and Cenke Xu for comments on the manuscript.

**Funding information** M.M. is supported by the National Science Foundation under Grant No. DMR-1847861. The work of A.K. was supported by the National Science Foundation

Graduate Research Fellowship under Grant No. 1745302. AK also acknowledges support from the Paul and Daisy Soros Fellowship and the Barry M. Goldwater Scholarship Foundation.

# A   Computation of the $\lambda$ Propagator

We now solve Eq. (39) for the $\lambda$ propagator. We again split our analysis into two half-planes. Let $K_{11}(y,z)$ correspond to when $y$ and $z$ are on the same half-plane, and let $K_{12}(y,z)$ correspond to when $y$ and $z$ are on opposite half-planes. Then, the system of equations we must solve are

$$\int_{\mathbb{R}^{3+}} d^3y \left\langle \lambda^1(x)\lambda^1(y) \right\rangle_c K_{11}(y,z) + \left\langle \lambda^1(x)\lambda^2(y) \right\rangle_c K_{12}(y,z) = \delta^3(x-z), \tag{A.157}$$

$$\int_{\mathbb{R}^{3+}} d^3y \left\langle \lambda^1(x)\lambda^2(y) \right\rangle_c K_{11}(y,z) + \left\langle \lambda^1(x)\lambda^1(y) \right\rangle_c K_{12}(y,z) = 0.$$

We define

$$\left\langle \lambda^1(x)\lambda^1(x') \right\rangle_c = \frac{2}{N} H_{11}(x,x') = \frac{2}{N} \frac{h_{11}(v)}{(4x^3 x'^3)^2},$$

$$\left\langle \lambda^1(x)\lambda^2(x') \right\rangle_c = \frac{2}{N} H_{12}(x,x') = \frac{2}{N} \frac{h_{12}(v)}{(4x^3 x'^3)^2}. \tag{A.158}$$

Then, we equivalently have to solve

$$\int_{\mathbb{R}^{3+}} d^3x \, H_{11}(x_1,x)G_{11}(x,x_2)^2 + H_{12}(x_1,x)G_{12}(x,x_2)^2 = \delta^3(x_1-x_2), \tag{A.159}$$

$$\int_{\mathbb{R}^{3+}} d^3x \, H_{11}(x_1,x)G_{12}(x,x_2)^2 + H_{12}(x_1,x)G_{11}(x,x_2)^2 = 0,$$

with $G_{11}$ and $G_{12}$ defined as in Eq. (23). We follow the method of Ref. [24]. Let

$$G^2_{11/12}(x,x') = \frac{\bar{g}_{11/12}(v)}{4x^3 x'^3}, \quad \bar{g}_{11/12}(v) = \frac{1}{16\pi^2} \frac{(v+\sqrt{v^2-1})^{2\mu} \pm 1 + (v+\sqrt{v^2-1})^{-2\mu}}{v^2-1}. \tag{A.160}$$

Furthermore, let

$$\hat{g}_{11/12}(\rho) = \frac{\pi}{2} \int_{2\rho+1}^{\infty} \bar{g}_{11/12}(v)dv, \qquad \hat{h}_{11/12}(\rho) = \frac{\pi}{2} \int_{2\rho+1}^{\infty} h_{11/12}(v)dv, \tag{A.161}$$

and $\rho_i = \frac{(x^3 - x_i^3)^2}{4x^3 x_i^3}$. Then, Eq. (A.159) reduces to

$$\int_0^{\infty} \frac{dx}{x} \hat{g}_{11}(\rho_1)\hat{h}_{11}(\rho_2) + \hat{g}_{12}(\rho_1)\hat{h}_{12}(\rho_2) = 4x_1^3 \delta\left(x_1^3 - x_2^3\right), \tag{A.162}$$

$$\int_0^{\infty} \frac{dx}{x} \hat{g}_{11}(\rho_1)\hat{h}_{12}(\rho_2) + \hat{g}_{12}(\rho_1)\hat{h}_{11}(\rho_2) = 0.$$

Finally, we define $F\{f\}(k)$ as the Fourier transform of $f(\sinh^2\theta)$. Then, Eq. (A.162) becomes

$$F\{\hat{g}_{11}\}(k)F\{\hat{h}_{11}\}(k) + F\{\hat{g}_{12}\}(k)F\{\hat{h}_{12}\}(k) = 1,$$

$$F\{\hat{g}_{12}\}(k)F\{\hat{h}_{11}\}(k) + F\{\hat{g}_{11}\}(k)F\{\hat{h}_{12}\}(k) = 0. \tag{A.163}$$

Thus, solving this equation for $F\{\hat{h}_{11/12}\}(k)$ allows us to compute $h_{11/12}(v)$.

In carrying out this procedure, we find that

$$\hat{g}_{11/12}(\rho) = \frac{1}{16\pi(1-2\mu)}z^{2\mu-1}F(1,1/2-\mu,3/2-\mu,1/z^2) \pm \frac{1}{8\pi}\text{arctanh}(1/z)+$$
$$\frac{1}{16\pi(1+2\mu)}z^{-2\mu-1}F(1,1/2+\mu,3/2+\mu,1/z^2), \quad \text{(A.164)}$$

where $z = 2\rho + 1 + \sqrt{(2\rho+1)^2 - 1}$. Then,

$$F\{\hat{g}_{11/12}\}(k) = \frac{\sinh(k\pi/2)}{16k(\cos(2\mu\pi)+\cosh(k\pi/2))} \pm \frac{1}{16k}\tanh(k\pi/4). \quad \text{(A.165)}$$

We note that strictly speaking the integral (A.161) for $\hat{g}_{11/12}(\rho)$ is only convergent for $\mu < 1/2$. Also, the Fourier transform (A.165) of (A.164) only exists for $\mu < 1/2$. We analytically continue Eq. (A.165) to $\mu > 1/2$. Notice that the result is invariant under $\mu \to 1-\mu$. Now solving Eqs. (A.163),

$$F\{\hat{h}_{11}\}(k) = 4k\,\text{csch}\,(k\pi/2)(1+\cos(2\mu\pi)+2\cosh(k\pi/2)),$$
$$F\{\hat{h}_{12}\}(k) = 4k\,\text{csch}(k\pi/2)(\cos(2\mu\pi)-1), \quad \text{(A.166)}$$

$$\hat{h}_{11}(\rho) = \frac{-8(\rho\sin^2(\mu\pi)+1)}{\rho(1+\rho)\pi}, \quad \hat{h}_{12}(\rho) = \frac{-8\sin^2(\mu\pi)}{(1+\rho)\pi}. \quad \text{(A.167)}$$

After taking a derivative of Eq. (A.161) with respect to $\rho$, we find

$$\left\langle \lambda^1(x)\lambda^1(x')\right\rangle_c = \frac{2}{(4x^3x'^3)^2N}h_{11}(v), \quad h_{11}(v) = \frac{32\cos^2(\mu\pi)}{(v+1)^2\pi^2} - \frac{32}{(v-1)^2\pi^2}, \quad \text{(A.168)}$$

$$\left\langle \lambda^1(x)\lambda^2(x')\right\rangle_c = \frac{2}{(4x^3x'^3)^2N}h_{12}(v), \quad h_{12}(v) = -\frac{32\sin^2(\mu\pi)}{(v+1)^2\pi^2}. \quad \text{(A.169)}$$

Again, the resulting $\lambda$ two-point function is invariant under $\mu \to 1-\mu$.

# B  Evaluation of Fig. 8(b) at Coincident Points

## B.1  Conformal Contribution

To compute Fig. 8(b) at coincident points, we evaluate the diagram for two points on the same side of the interface:

$$G_{11}^{(b)}(x,y) = -\frac{2}{N}\int d^3w\,d^3z\,G_{11}(x,w)G_{11}(w,z)H_{11}(w,z)G_{11}(z,y)+ \quad \text{(B.1)}$$
$$G_{11}(x,w)G_{12}(w,z)H_{12}(w,z)G_{12}(z,y)+$$
$$G_{12}(x,w)G_{12}(w,z)H_{12}(w,z)G_{11}(z,y)+$$
$$G_{12}(x,w)G_{11}(w,z)H_{11}(w,z)G_{12}(z,y).$$

Here, we use the definition of $H_{11/12}$ from Eq. (A.158). We now define the differential operators

$$\mathcal{L}_x = \left(-\mathbf{\nabla}_x^2 + \frac{\mu^2 - 1/4}{(x^3)^2}\right), \quad \mathcal{L}_y = \left(-\mathbf{\nabla}_y^2 + \frac{\mu^2 - 1/4}{(y^3)^2}\right). \quad \text{(B.2)}$$

Then,

$$\mathcal{L}_x\mathcal{L}_yG_{11}^{(b)}(x,y) = -\frac{2}{N}G_{11}(x,y)H_{11}(x,y). \quad \text{(B.3)}$$

To the order we are interested, the diagram has the form

$$G_{11}^{(b)}(x,y) = \frac{g_{11}^{(b)}(v)}{\sqrt{x^3 y^3}} + G_{\text{nconf.}}^{(b)}(x,y), \tag{B.4}$$

where the latter term arises because of the $UV$ divergence in $-2/N G_{11}(x,y) H_{11}(x,y)$. As we check in App. B.2, the latter term vanishes under the application of $\mathcal{L}_x \mathcal{L}_y$. Now, let

$$\mathcal{D} = \mu^2 - 1 - 3v \frac{\mathrm{d}}{\mathrm{d}v} - (v^2 - 1) \frac{\mathrm{d}^2}{\mathrm{d}v^2}. \tag{B.5}$$

Then,

$$\mathcal{L}_x \mathcal{L}_y G_{11}^{(b)}(x,x') = \frac{1}{(x^3)^{5/2}(x'^3)^{5/2}} \mathcal{D}^2 g_{11}^{(b)}. \tag{B.6}$$

Then, Eq. (B.3) reduces to

$$\mathcal{D}^2 g_{11}^{(b)}(v) = -\frac{(v + \sqrt{v^2 - 1})^\mu + (v + \sqrt{v^2 - 1})^{-\mu}}{2\pi^3 N \sqrt{v^2 - 1}} \left( \frac{\cos^2(\mu\pi)}{(v+1)^2} - \frac{1}{(v-1)^2} \right). \tag{B.7}$$

For simplicity in future notation, we define $g_{11}^{(b)}(v) = -\frac{1}{2\pi^3 N} c(v)$.

For later convenience, we split up our analysis into symmetric and antisymmetric channels. We do this as follows. Note that

$$\mathcal{L}_x \mathcal{L}_y G_{12}^{(b)}(x,y) = -\frac{2}{N} G_{12}(x,y) H_{12}(x,y). \tag{B.8}$$

Then, if we define (note the normalization)

$$G_{S/A}^{(b)}(x,y) = \frac{1}{2} G_{11}^{(b)} \pm \frac{1}{2} G_{12}^{(b)}, \tag{B.9}$$

and we define $g_{S/A}^{(b)}$ analogously to our definition of $g_{11}^{(b)}$, we find that

$$\mathcal{D}^2 g_S^{(b)}(v) = -\frac{1}{4\pi^3 N} s_S(v) \left( \frac{\cos(2\mu\pi)}{(v+1)^2} - \frac{1}{(v-1)^2} \right) - \frac{1}{4\pi^3 N} s_A(v) \left( \frac{1}{(v+1)^2} - \frac{1}{(v-1)^2} \right), \tag{B.10}$$

$$\mathcal{D}^2 g_A^{(b)}(v) = -\frac{1}{4\pi^3 N} s_A(v) \left( \frac{\cos(2\mu\pi)}{(v+1)^2} - \frac{1}{(v-1)^2} \right) - \frac{1}{4\pi^3 N} s_S(v) \left( \frac{1}{(v+1)^2} - \frac{1}{(v-1)^2} \right), \tag{B.11}$$

where $s_S(v)$ and $s_A(v)$ are defined in Eq. (29). If we define $c_S(v)$ and $c_A(v)$ analogously to $c(v)$, we find

$$c(v) = c_S(v) + c_A(v). \tag{B.12}$$

There are four independent homogenous solutions to $\mathcal{D}^2 f = 0$: two symmetric solutions $s_S(v)$ and $-\frac{s_S(v)\cosh^{-1}(v)}{2\mu}$, and two antisymmetric solutions $s_A(v)$ and $\frac{s_A(v)\cosh^{-1}(v)}{2\mu}$. The advantage of splitting our analysis into symmetric and antisymmetric channels is that only the symmetric homogenous solutions are allowed to enter $c_S(v)$ and only the antisymmetric homogenous solutions are allowed to enter $c_A(v)$. One way to argue this is by noting that the spectrum of boundary operators in the symmetric/antisymmetric channels should not be drastically changed by $1/N$ corrections. Our strategy is thus to compute the inhomogenous solutions for the $c_{S/A}$ differential equations, to use boundary conditions to constrain $c_S(v)$ and $c_A(v)$, and to use these expressions to find $G_{11, \text{conf.}}^{(b)}(x,x) - G_{11, \text{bulk}}^{(b)}(x,x)$ (where the latter term represents the bulk divergences).

### B.1.1 Inhomogenous Solution

We use the method of variation of constants to compute the inhomogenous solution to $c_S$ and $c_A$. Note that $s_S(v)$ and $s_A(v)$ are the two homogenous solutions to the differential operator $\mathcal{D}$. Let

$$\mathcal{D}c_{S/A} = c_{S/A}^{\text{int}}. \tag{B.13}$$

Let us label the right hand side of the symmetric and antisymmetric differential equations for $c_S(v)$ and $c_A(v)$ as $f_{S/A}(v)$. We first start with the inhomogenous symmetric solution.

$$c_S^{\text{int}}(v) = s_S(v) \int_v^\infty \frac{f_S(v')s_A(v')\sqrt{v'^2-1}}{2\mu}\,dv' - s_A(v) \int_v^\infty \frac{f_S(v')s_S(v')\sqrt{v'^2-1}}{2\mu}\,dv'. \tag{B.14}$$

We expand this as

$$c_S^{\text{int}}(v) = \frac{s_S(v)}{4\mu} \int_v^\infty \frac{dv'}{\sqrt{v'^2-1}}\left(\frac{\cos(2\mu\pi)}{(v'+1)^2} - \frac{1}{(v'-1)^2}\right) + \tag{B.15}$$

$$\frac{s_S(v)}{4\mu} \int_v^\infty \frac{dv'(v'-\sqrt{v'^2-1})^{2\mu}}{\sqrt{v'^2-1}}\left(\frac{1}{(v'+1)^2} - \frac{1}{(v'-1)^2}\right) -$$

$$\frac{s_A(v)}{4\mu} \int_v^\infty \frac{dv'}{\sqrt{v'^2-1}}\left(\frac{1}{(v'+1)^2} - \frac{1}{(v'-1)^2}\right) -$$

$$\frac{s_A(v)}{4\mu} \int_v^\infty \frac{dv'(v'+\sqrt{v'^2-1})^{2\mu}}{\sqrt{v'^2-1}}\left(\frac{\cos(2\mu\pi)}{(v'+1)^2} - \frac{1}{(v'-1)^2}\right).$$

We substitute $v = \cosh\theta$ again and get

$$c_S^{\text{int}}(\cosh\theta) = \frac{e^{\mu\theta}}{4\mu\sinh\theta} \int_\theta^\infty d\alpha\left(\frac{\cos(2\mu\pi)}{(\cosh\alpha+1)^2} - \frac{1}{(\cosh\alpha-1)^2}\right) + \tag{B.16}$$

$$\frac{e^{\mu\theta}}{4\mu\sinh\theta} \int_\theta^\infty d\alpha\, e^{-2\mu\alpha}\left(\frac{1}{(\cosh\alpha+1)^2} - \frac{1}{(\cosh\alpha-1)^2}\right) -$$

$$\frac{e^{-\mu\theta}}{4\mu\sinh\theta} \int_\theta^\infty d\alpha\left(\frac{1}{(\cosh\alpha+1)^2} - \frac{1}{(\cosh\alpha-1)^2}\right) -$$

$$\frac{e^{-\mu\theta}}{4\mu\sinh\theta} \int_\theta^\infty d\alpha\, e^{2\mu\alpha}\left(\frac{\cos(2\mu\pi)}{(\cosh\alpha+1)^2} - \frac{1}{(\cosh\alpha-1)^2}\right).$$

Let us define

$$I_\pm(q,\theta) = \int_\theta^\infty d\alpha\,\frac{e^{q\alpha}}{(\cosh\alpha\pm1)^2}, \qquad \tilde{I}_\pm(q,\theta) = \int_\theta^\infty d\alpha\,\frac{\alpha e^{q\alpha}}{(\cosh\alpha\pm1)^2}. \tag{B.17}$$

The closed forms of these integrals are in Appendix F. Then,

$$c_S^{\text{int}}(\cosh\theta) = \frac{e^{\mu\theta}}{4\mu\sinh\theta}\left(\cos(2\mu\pi)I_+(0,\theta) - I_-(0,\theta) + I_+(-2\mu,\theta) - I_-(-2\mu,\theta)\right) \tag{B.18}$$

$$-\frac{e^{-\mu\theta}}{4\mu\sinh\theta}\left(I_+(0,\theta) - I_-(0,\theta) + \cos(2\mu\pi)I_+(2\mu,\theta) - I_-(2\mu,\theta)\right).$$

The inhomogenous symmetric solution is

$$c_S^{\text{inhomog.}}(v) = s_S(v) \int_v^\infty \frac{c_S^{\text{int}}(v')s_A(v')\sqrt{v'^2-1}}{2\mu}\,dv' - s_A(v) \int_v^\infty \frac{c_S^{\text{int}}(v')s_S(v')\sqrt{v'^2-1}}{2\mu}\,dv'. \tag{B.19}$$

We compute this function by integrating by parts. Namely, we use that

$$\int_\theta^\infty d\beta f_2(\beta) \int_\beta^\infty d\alpha f_1(\alpha) = \int_\theta^\infty d\alpha f_1(\alpha) \int_\theta^\alpha d\beta f_2(\beta). \tag{B.20}$$

Then,

$$c_S^{\text{inhomog.}}(\cosh\theta) = -\frac{\operatorname{csch}\theta}{8\mu^3}\Big[e^{-\mu\theta}(\mu\theta+1)(I_+(0,\theta)+\cos(2\mu\pi)I_+(2\mu,\theta))$$
$$+ e^{\mu\theta}(\mu\theta-1)(I_+(-2\mu,\theta)+\cos(2\mu\pi)I_+(0,\theta))\Big] \tag{B.21}$$

$$+ \frac{\operatorname{csch}\theta}{8\mu^3}\Big[e^{-\mu\theta}(\mu\theta+1)(I_-(0,\theta)+I_-(2\mu,\theta))$$
$$+ e^{\mu\theta}(\mu\theta-1)(I_-(-2\mu,\theta)+I_-(0,\theta))\Big] \tag{B.22}$$

$$+ \frac{\operatorname{csch}\theta}{8\mu^2}\Big[e^{-\mu\theta}(\tilde{I}_+(0,\theta)+\cos(2\mu\pi)\tilde{I}_+(2\mu,\theta))$$
$$+ e^{\mu\theta}(\tilde{I}_+(-2\mu,\theta)+\cos(2\mu\pi)\tilde{I}_+(0,\theta))\Big] \tag{B.23}$$

$$- \frac{\operatorname{csch}\theta}{8\mu^2}\Big[e^{-\mu\theta}(\tilde{I}_-(0,\theta)+\tilde{I}_-(2\mu,\theta)) + e^{\mu\theta}(\tilde{I}_+(-2\mu,\theta)+\tilde{I}_-(0,\theta))\Big].$$

We analogously compute

$$c_A^{\text{int}}(\cosh\theta) = \frac{e^{\mu\theta}}{4\mu\sinh\theta}(I_+(0,\theta)-I_-(0,\theta)+\cos(2\mu\pi)I_+(-2\mu,\theta)-I_-(-2\mu,\theta))- \tag{B.24}$$
$$\frac{e^{-\mu\theta}}{4\mu\sinh\theta}(\cos(2\mu\pi)I_+(0,\theta)-I_-(0,\theta)+I_+(2\mu,\theta)-I_-(2\mu,\theta)),$$

$$c_A^{\text{inhomog.}}(\cosh\theta) = -\frac{\operatorname{csch}\theta}{8\mu^3}\Big[e^{-\mu\theta}(\mu\theta+1)(\cos(2\mu\pi)I_+(0,\theta)+I_+(2\mu,\theta))$$
$$+ e^{\mu\theta}(\mu\theta-1)(\cos(2\mu\pi)I_+(-2\mu,\theta)+I_+(0,\theta))\Big] \tag{B.25}$$

$$+ \frac{\operatorname{csch}\theta}{8\mu^3}\Big[e^{-\mu\theta}(\mu\theta+1)(I_-(0,\theta)+I_-(2\mu,\theta))$$
$$+ e^{\mu\theta}(\mu\theta-1)(I_-(-2\mu,\theta)+I_-(0,\theta))\Big] \tag{B.26}$$

$$+ \frac{\operatorname{csch}\theta}{8\mu^2}\Big[e^{-\mu\theta}(\cos(2\mu\pi)\tilde{I}_+(0,\theta)+\tilde{I}_+(2\mu,\theta))$$
$$+ e^{\mu\theta}(\cos(2\mu\pi)\tilde{I}_+(-2\mu,\theta)+\tilde{I}_+(0,\theta))\Big] \tag{B.27}$$

$$- \frac{\operatorname{csch}\theta}{8\mu^2}\Big[e^{-\mu\theta}(\tilde{I}_-(0,\theta)+\tilde{I}_-(2\mu,\theta)) + e^{\mu\theta}(\tilde{I}_+(-2\mu,\theta)+\tilde{I}_-(0,\theta))\Big].$$

We are after the value of $c(1)$ after subtracting off the bulk divergences. Thus, we expand $c(\cosh\theta) = c_S(\cosh\theta) + c_A(\cosh\theta)$ for $\theta \ll 1$ using App. F and extract the constant term:

$$c_S^{\text{inhomog.}}(1) + c_A^{\text{inhomog.}}(1)\Big|_{\text{subtracted}} =$$
$$\frac{\pi\sec^2(\mu\pi)\big(2\mu(4\mu^2-1)\pi(2+\cos(2\mu\pi)) + (12\mu^2-1)\sin(2\mu\pi)\big)}{24\mu}. \tag{B.28}$$

### B.1.2 Boundary Conditions

The full solutions to the differential equation are

$$c_S(v) = c_S^{\text{inhomog.}}(v) + C_1 s_s(v) - \frac{C_3 s_S(v)\cosh^{-1}(v)}{2\mu} \,, \tag{B.29}$$

$$c_A(v) = c_A^{\text{inhomog.}}(v) + C_2 s_A(v) + \frac{C_4 s_A(v)\cosh^{-1}(v)}{2\mu} \,.$$

We constrain the coefficients of the homogenous solutions using the following boundary conditions: (i) the difference between $c_S(v)$ and $c_A(v)$ does not have a pole at $v = 1$, (ii) the difference between $c_S(v)$ and $c_A(v)$ has no $\sqrt{v-1}$ term in its series expansion around $v = 1$. Boundary condition (i) arises because $G_{12}^{(b)} = G_S^{(b)} - G_A^{(b)}$, and at coincident points, $G_{12}^{(b)}$ must be nonsingular (the two points are on different half-planes). Boundary condition (ii) arises because the self-energy for $G_{12}^{(b)}$, i.e., $\mathcal{L}_x \mathcal{L}_y G_{12}^{(b)}(x,y)$ or the RHS of Eq. (B.8), has no delta function. These two boundary conditions are sufficient for computing $c(1)|_{\text{subtracted}}$, the value of $c(1)$ after subtracting off bulk divergences.

Boundary condition (i) directly gives us the value of $C_1 - C_2$. We can express $C_1 - C_2$ both in terms of integrals and explicitly – both forms are useful for this computation. Let $\Delta c^{\text{int}} = c_S^{\text{int}} - c_A^{\text{int}}$. Then,

$$\begin{aligned}
c^S(v) - c^A(v) = {}& s_S(v)\int_v^\infty \frac{\Delta c_{\text{int}}(v') s_A(v')\sqrt{v'^2-1}}{2\mu}\mathrm{d}v' - \\
& s_A(v)\int_v^\infty \frac{\Delta c_{\text{int}}(v') s_S(v')\sqrt{v'^2-1}}{2\mu}\mathrm{d}v' + \\
& C_1 s_S(v) - C_2 s_A(v) - \frac{C_3}{2\mu} s_S(v)\cosh^{-1}(v) - \frac{C_4}{2\mu} s_A(v)\cosh^{-1}(v)\,.
\end{aligned} \tag{B.30}$$

Now, note that

$$\Delta c^{\text{int}}(\cosh\theta) = \frac{e^{\mu\theta}\sin^2(\mu\pi)}{2\mu\sinh\theta}\left(I_+(-2\mu,\theta) - I_+(0,\theta)\right) + \frac{e^{-\mu\theta}\sin^2(\mu\pi)}{2\mu\sinh\theta}\left(I_+(0,\theta) - I_+(2\mu,\theta)\right)\,. \tag{B.31}$$

Importantly, per the expressions in App. F, $\Delta c^{\text{int}}(\cosh\theta)$ does not diverge for small $\theta$, and $s_{S/A}(v)\sqrt{v^2-1}$ does not diverge as $v \to 1$, so none of the integrands in Eq. (B.30) diverge as $v \to 1$. Thus, all possible divergences arise because both $s_S(v)$ and $s_A(v)$ are singular as $v \to 1$. Matching the singular behavior of the first two terms in Eq. (B.30) with the next two terms in Eq. (B.30) gives us that

$$C_1 - C_2 = \int_1^\infty \frac{\Delta c_{\text{int}}(v') s_-(v')\sqrt{v'^2-1}}{2\mu}\,\mathrm{d}v'\,, \tag{B.32}$$

where $s_-(v) = s_S(v) - s_A(v)$. We can also directly compute $C_1 - C_2$ using Eqs. (B.21) and (B.25) along with the series expansions in App. F:

$$C_1 - C_2 = \frac{-\sec^2(\mu\pi)(-1 + 4\mu^2(1-4\mu^2)\pi^2\cos(2\mu\pi) + \cos(4\mu\pi) + 2(\mu+4\mu^3)\pi\sin(2\mu\pi))}{48\mu^3}\,. \tag{B.33}$$

For boundary condition (ii), we need the $\sqrt{v-1}$ terms in the series expansions of the first four terms in Eq. (B.30). First note that

$$s_S(v) \approx \frac{1}{\sqrt{2(v-1)}} + \mu + \frac{(\mu^2-1/4)\sqrt{v-1}}{\sqrt{2}} + \cdots\,, \tag{B.34}$$

$$s_A(v) \approx \frac{1}{\sqrt{2(v-1)}} - \mu + \frac{(\mu^2 - 1/4)\sqrt{v-1}}{\sqrt{2}} + \cdots \tag{B.35}$$

We begin with the series expansion of the integrals. First, note that

$$\frac{1}{2\mu}\Delta c_{\text{int}}(v)s_A(v)\sqrt{v^2-1} \approx \frac{(2\mu\pi - 8\mu^3\pi - \sin(2\mu\pi))\tan(\mu\pi)}{12\mu^2\sqrt{2(v-1)}} + i_A(\mu) + \cdots, \tag{B.36}$$

where,

$$i_A(\mu) =$$
$$\frac{1}{12\mu}\Big(4\mu\pi(4\mu^2-1) + (2+\mu-4\mu^2+2\mu(4\mu^2-1)(-H(\mu-1/2)+H(\mu)))\sin(2\mu\pi)\Big)\tan(\mu\pi), \tag{B.37}$$

and $H$ is the harmonic number. Then, the contribution from

$$s_S(v)\int_v^\infty \frac{1}{2\mu}\Delta c_{\text{int}}(v')s_A(v')\sqrt{v'^2-1}\,dv' \tag{B.38}$$

to the $\sqrt{v-1}$ term is

$$-\frac{i_A(\mu)\sqrt{v-1}}{\sqrt{2}} - \frac{(\mu^2-1/4)\sqrt{v-1}}{\sqrt{2}}\int_1^\infty \frac{1}{2\mu}\Delta c_{\text{int}}(v')s_A(v')\sqrt{v'^2-1}\,dv' \tag{B.39}$$
$$-\frac{2\tan(\mu\pi)(2\mu\pi-8\mu^3\pi-\sin(2\mu\pi))\sqrt{v-1}}{12\sqrt{2}\mu}.$$

Likewise,

$$\frac{1}{2\mu}\Delta c_{\text{int}}(v)s_S(v)\sqrt{v^2-1} \approx \frac{(2\mu\pi - 8\mu^3\pi - \sin(2\mu\pi))\tan(\mu\pi)}{12\mu^2\sqrt{2(v-1)}} + i_S(\mu) + \cdots, \tag{B.40}$$

where

$$i_S(\mu) = \frac{1}{12\mu}\Big(-4\mu\pi(4\mu^2-1)+$$
$$\big(-2+\mu+4\mu^2+2\mu(4\mu^2-1)\big(-H(-\mu-1/2)+H(-\mu)\big)\big)\sin(2\mu\pi)\Big)\tan(\mu\pi). \tag{B.41}$$

Then, the contribution from

$$-s_A(v)\int_v^\infty \frac{1}{2\mu}\Delta c_{\text{int}}(v')s_S(v')\sqrt{v'^2-1}\,dv' \tag{B.42}$$

to the $\sqrt{v-1}$ term is

$$\frac{i_S(\mu)\sqrt{v-1}}{\sqrt{2}} - \frac{(\mu^2-1/4)\sqrt{v-1}}{\sqrt{2}}\int_1^\infty \frac{1}{2\mu}\Delta c_{\text{int}}(v')s_S(v')\sqrt{v'^2-1}\,dv' \tag{B.43}$$
$$-\frac{2\tan(\mu\pi)(2\mu\pi-8\mu^3\pi-\sin(2\mu\pi))\sqrt{v-1}}{12\sqrt{2}\mu}.$$

Summing these two contributions and simplifying gives

$$-\frac{(\mu^2-1/4)\sqrt{v-1}}{\sqrt{2}}\int_1^\infty \frac{1}{2\mu}\Delta c_{\text{int}}(v')s_-(v')\sqrt{v'^2-1}\,dv'$$
$$-\frac{\tan(\mu\pi)(2\mu\pi-8\mu^3\pi-\sin(2\mu\pi))\sqrt{v-1}}{6\sqrt{2}\mu}. \tag{B.44}$$

Per Eq. (B.32), the contribution from $C_1 s_S(v) - C_2 s_A(v)$ is

$$\frac{(\mu^2 - 1/4)\sqrt{v-1}}{\sqrt{2}} \int_1^\infty \frac{\Delta c_{\text{int}}(v') s_-(v') \sqrt{v'^2 - 1}}{2\mu} \mathrm{d}v'. \tag{B.45}$$

Then, the two integrals cancel, and to eliminate the coefficient of $\sqrt{v-1}$ we require

$$C_3 - C_4 = -\frac{\tan(\mu\pi)(2\mu\pi - 8\mu^3\pi - \sin(2\mu\pi))}{6\mu}. \tag{B.46}$$

### B.1.3 Final Result

Using that $c(v) = c_S(v) + c_A(v)$, we arrive at

$$c(1)|_{\text{subtracted}} = c_S^{\text{inhomog.}}(1)\Big|_{\text{subtracted}} + c_A^{\text{inhomog.}}(1)\Big|_{\text{subtracted}} + \mu(C_1 - C_2) - \frac{C_3 - C_4}{2\mu}. \tag{B.47}$$

This simplifies to

$$c(1)|_{\text{subtracted}} = \frac{4\pi^2}{3}(\mu^2 - 1/4). \tag{B.48}$$

Thus,

$$G_{11,\text{ conf.}}^{(b)}(x,x) = -\frac{2}{3\pi N}(\mu^2 - 1/4). \tag{B.49}$$

### B.2 Nonconformal Contribution

The nonconformal contribution arises from a *UV* divergence in the self-energy of the diagram in Fig. 8(b). Thus, we are interested in the terms in Eq. (B.1) where $w$ and $z$ are on the same side of the interface:

$$G_{ss,11}^{(b)}(x,y) = -\frac{2}{N} \int_{|w-z|\geq a} \mathrm{d}^3w\, \mathrm{d}^3z$$
$$[G_{11}(w,z)H_{11}(w,z)][G_{11}(x,w)G_{11}(z,y) + G_{12}(x,w)G_{12}(z,y)]. \tag{B.50}$$

Here, $a$ is a lattice cutoff. We follow the method of Ref. [1] to compute the effect of the lattice cutoff. Note that $G_{11}(w,z)H_{11}(w,z)$ has the form of a two-point function of a conformal scalar of dimension $5/2$. Then, we know that under a small conformal transformation $x^\mu \to x^\mu + \epsilon^\mu(x)$,

$$\delta_\epsilon G_{ss,11}^{(b)}(x,y) \approx -\frac{2}{N} \int \mathrm{d}^3w\, \mathrm{d}^3z \frac{(w-z)\cdot(\epsilon(w) - \epsilon(z))}{|w-z|} \delta(|w-z| - a)(G_{11}(w,z)H_{11}(w,z)) \tag{B.51}$$

$$\cdot (G_{11}(x,w)G_{11}(z,y) + G_{12}(x,w)G_{12}(z,y)).$$

Now, recall that if $s = z - w$,

$$G_{11}(w,z) = \frac{1}{8\pi\sqrt{w^3 z^3}\sqrt{v^2 - 1}}\left((v + \sqrt{v^2 - 1})^\mu - (v + \sqrt{v^2 - 1})^{-\mu}\right), \qquad v = \frac{s^2}{2w^3 z^3} + 1, \tag{B.52}$$

$$H_{11}(w,z) = \frac{2}{\pi^2(z^3 w^3)^2}\left(\frac{\cos^2(\mu\pi)}{(v+1)^2} - \frac{1}{(v-1)^2}\right). \tag{B.53}$$

Then, we can expand the self-energy and propagators in the small $s$ limit:

$$G_{11}H_{11} \approx -\frac{1}{4\pi^3}\left(\frac{8}{s^5} + \frac{4\mu^2-1}{s^3(w^3)^2} + \cdots\right), \tag{B.54}$$

$$G_{11/12}(z,y) = (1 + s^\mu\partial_\mu^w + (1/2)s^\mu s^\nu\partial_\mu^w\partial_\nu^w + \cdots)G_{11/12}(w,y). \tag{B.55}$$

We now see how $\delta_\epsilon G_{11}^{(b)}$ transforms under the two types of conformal transformations: scale transformations and special conformal transformations. First, consider a scale transformation: $\epsilon^\mu(x) = \epsilon x^\mu$. Then,

$$\delta_\epsilon G_{11}^{(b)}(x,y) \approx -\frac{\epsilon}{2\pi^3 N}\int \mathrm{d}^3w\left(\frac{8}{a^2} + \frac{4\mu^2-1}{(w^3)^2}\right)$$
$$\int \sin\theta\,\mathrm{d}\theta\,\mathrm{d}\phi(G_{11}(x,w)G_{11}(z,y) + G_{12}(x,w)G_{12}(z,y)) \tag{B.56}$$
$$\approx -\frac{2\epsilon}{\pi^2 N}\int \mathrm{d}^3w\left(\frac{8}{a^2} + \frac{4\mu^2-1}{(w^3)^2}\right)\left[G_{11}(x,w)\left(1 + \frac{1}{6}a^2\partial_w^2 G_{11}(w,y)\right)\right.$$
$$\left. + G_{12}(x,w)\left(1 + \frac{1}{6}a^2\partial_w^2 G_{12}(w,y)\right)\right]. \tag{B.57}$$

We drop a term that diverges as $a^{-2}$ because it is canceled by a shift in the expectation value, $\langle\delta\lambda\rangle$, at the critical point. The constant term is

$$\delta_\epsilon G_{11}^{(b)}(x,y)\Big|_{\mathrm{const}} \approx -\frac{2\epsilon}{\pi^2 N}\int \mathrm{d}^3w\left[\frac{4}{3}\left(G_+(x,w)\partial_w^2 G_+(x,w) + G_-(x,w)\partial_w^2 G_-(y,w)\right)\right. \tag{B.58}$$
$$\left. + \frac{4\mu^2-1}{(w^3)^2}\left(G_+(x,w)G_+(w,y) + G_-(x,w)G_-(w,y)\right)\right].$$

Finally, we have that

$$\left[-\partial_w^2 + \frac{\mu^2-1/4}{(w^3)^2}\right]G_{11}(w,x) = \delta^3(w,x), \qquad \left[-\partial_w^2 + \frac{\mu^2-1/4}{(w^3)^2}\right]G_{12}(w,x) = 0. \tag{B.59}$$

Then,

$$\delta_\epsilon G_{11}^{(b)}(x,y)\Big|_{\mathrm{const}} \approx \frac{8\epsilon}{3\pi^2 N}G_{11}(x,y)$$
$$- \frac{8\epsilon}{3\pi^2 N}\int \mathrm{d}^3w\left[\frac{4\mu^2-1}{(w^3)^2}\left(G_{11}(x,w)G_{11}(w,y) + G_{12}(x,w)G_{12}(w,y)\right)\right]. \tag{B.60}$$

Now, consider a special conformal transformation: $\epsilon^\mu = b^\mu x^2 - 2(b\cdot x)x^\mu$. Now,

$$\frac{(w-z)\cdot(\epsilon(w)-\epsilon(z))}{|w-z|} = 2(b\cdot w)s + (b\cdot s)s. \tag{B.61}$$

Substituting gives

$$\delta_\epsilon G_{11}^{(b)}(x,y) \approx \frac{1}{2\pi^3 N}\int \mathrm{d}^3w\left(\frac{8}{a^2} + \frac{4\mu^2-1}{(w^3)^2}\right)\int \sin\theta\,\mathrm{d}\theta\,\mathrm{d}\phi(2b\cdot w + b\cdot s)$$
$$\cdot(G_{11}(x,w)G_{11}(z,y) + G_{12}(x,w)G_{12}(z,y)). \tag{B.62}$$

The first term in $(2b \cdot w + b \cdot s)$ acts a constant, whereas the second term fixes $b$ parallel to the first derivative in the Taylor expansion of $G_{11/12}(z, y)$. Then,

$$
\delta_\epsilon G_{11}^{(b)}(x, y) \approx \frac{4}{\pi^2 N} \int d^3 w \, G_{11}(x, w) \left[ (b \cdot w)\left( \frac{4}{3}\partial_w^2 + \frac{4\mu^2 - 1}{(w^3)^2} \right) + \frac{4}{3} b^\mu \partial_\mu^w \right] G_{11}(w, y) +
$$
$$
G_{12}(x, w) \left[ (b \cdot w)\left( \frac{4}{3}\partial_w^2 + \frac{4\mu^2 - 1}{(w^3)^2} \right) + \frac{4}{3} b^\mu \partial_\mu^w \right] G_{12}(w, y).
$$
(B.63)

Then,

$$
\delta_\epsilon G_{11}^{(b)}(x, y) \approx -\frac{8}{3N\pi^2} b \cdot (x + y) G_+(x, y) + \frac{16(4\mu^2 - 1)}{3\pi^2 N}
$$
$$
\times \int d^3 w \left[ \frac{b \cdot w}{(w^3)^2} \left( G_{11}(x, w) G_{11}(w, y) + G_{12}(x, w) G_{12}(w, y) \right) \right].
$$
(B.64)

Thus, we need an expression for $G_{\text{nconf}}^{(b)}(x, y)$ that vanishes under $\mathcal{L}_x \mathcal{L}_\dagger$ and transforms as described above for both scale transformations and special conformal transformations. The below expression satisfies both constraints (it vanishes under $\mathcal{L}_x \mathcal{L}_y$ up to contact terms):

$$
G_{\text{nconf}}^{(b)}(x, y) = \frac{32(\mu^2 - 1/4)}{3N\pi^2} \int d^3 w \, \frac{\log(\Lambda' w^3)}{(w^3)^2} \left( G_{11}(x, w) G_{11}(w, y) + G_{12}(x, w) G_{12}(w, y) \right)
$$
$$
- \frac{4}{3N\pi^2} \log\left( 4x^3 y^3 \Lambda''^2 \right) G_{11}(x, y).
$$
(B.65)

Here, $\Lambda'$ and $\Lambda''$ are two UV cutoffs that are lattice-dependent. They are not necessarily equivalent, but they both inversely scale with the lattice spacing.

## C  Perturbation theory around the ordinary fixed point

In this section we perform perturbation theory in the coupling $u$, Eq. (4), around the ordinary fixed point for the interface. Our goal is to match $u$ to $\mu$, Eq. (25), in the $\mu, u \to 0$ limit. We set $N = \infty$ throughout. We consider the anomalous dimension of the boundary operators $\hat{\phi}_S$, $\hat{\phi}_A$ to first order in $u$ and match it to $\hat{\Delta}_S = 1 - \mu$, $\hat{\Delta}_A = 1 + \mu$. We have $\langle \hat{\phi}_a^1(r) \hat{\phi}_b^1(0) \rangle_{\text{ord}} = \langle \hat{\phi}_a^2(r) \hat{\phi}_b^2(0) \rangle_{\text{ord}} = \frac{\delta_{ab}}{r^2}$, $\langle \hat{\phi}_a^1(r) \hat{\phi}_b^2(0) \rangle_{\text{ord}} = 0$. Then to first order in $u$,

$$
\langle \hat{\phi}_a^1(x) \hat{\phi}_b^2(y) \rangle = -u \int d^2 r \langle \hat{\phi}_a^1(x) \hat{\phi}_c^1(r) \rangle_{\text{ord}} \langle \hat{\phi}_b^2(y) \hat{\phi}_c^2(r) \rangle_{\text{ord}}
$$
$$
= -u\delta_{ab} \int d^2 r \frac{1}{(r - x)^2 (r - y)^2}
$$
$$
= -4\pi u\delta_{ab} \frac{\log \Lambda |x - y|}{|x - y|^2}.
$$
(C.1)

The first order correction in $u$ to $\langle \hat{\phi}_a^1(x) \hat{\phi}_b^1(y) \rangle$, $\langle \hat{\phi}_a^2(x) \hat{\phi}_b^2(y) \rangle$ is zero. Thus,

$$
\langle \hat{\phi}_{S/A,a}(x) \hat{\phi}_{S/A,b}(y) \rangle \approx \frac{\delta_{ab}}{|x - y|^2} \left( 1 \mp 4\pi u \log \Lambda |x - y| \right),
$$
(C.2)

from which we read off $\hat{\Delta}_S = 1 + 2\pi u$, $\hat{\Delta}_A = 1 - 2\pi u$. Thus, to leading order $\mu = -2\pi u$.

# D  Free energy on $HS^3$

For a single scalar in the background of $\langle i\lambda \rangle$ in Eq. (70) the free energy on $HS^3$ is

$$F_{HS^3} = -\log Z = \frac{1}{2}\mathrm{Tr}\log\left(-\Delta + \langle i\lambda \rangle + \frac{3}{4R^2}\right). \tag{D.1}$$

In this appendix, we repeat for completeness the calculation of the trace (D.1) presented in Ref. [27].

We need to know the spectrum of $-\Delta + \langle i\lambda \rangle$. We have

$$-\Delta + \langle i\lambda \rangle = -\partial_\alpha^2 - 2\cot\alpha\,\partial_\alpha + \csc^2\alpha(-\Delta_{S^2}) + (\mu^2 - 1/4)\sec^2\alpha, \tag{D.2}$$

where we have temporarily set the radius $R$ to one. Going to sectors of fixed angular momentum $\ell$, $-\Delta_{S^2} \to \ell(\ell+1)$ we find eigenfunctions

$$(-\Delta + \langle i\lambda \rangle)\phi_\ell^\pm = (-1 + k^2)\phi_\ell^\pm, \qquad k > 0, \tag{D.3}$$

$$\phi_\ell^\pm(\alpha) = (\cos\alpha)^{\frac{1}{2}\pm\mu}(\sin\alpha)^\ell\, {}_2F_1\left(\frac{3}{4} - \frac{k}{2} + \frac{\ell}{2} \pm \frac{\mu}{2}, \frac{3}{4} + \frac{k}{2} + \frac{\ell}{2} \pm \frac{\mu}{2}, 1 \pm \mu, \cos^2\alpha\right). \tag{D.4}$$

Note that as $\alpha \to \frac{\pi}{2}$, $\phi_\ell^\pm(\alpha) \to (\frac{\pi}{2} - \alpha)^{\frac{1}{2}\pm\mu}$. The "+" solution gives rise to the boundary fixed point with $\hat{\Delta}_\phi = 1 + \mu$, and the "−" solution gives rise to the boundary fixed point with $\hat{\Delta}_\phi = 1 - \mu$, which we have encountered in section 3. Note also that $\phi^-$ can be obtained from $\phi^+$ by substituting $\mu \to -\mu$. Thus, we work with $\phi^+$ throughout and use $\mu > 0$ for $\hat{\Delta}_\phi = 1 + |\mu|$ and $\mu < 0$ for $\hat{\Delta}_\phi = 1 - |\mu|$.

In order for $\phi^+$ to be finite at $\alpha = 0$, the first index of the hypergeometric function must be equal $-n$, $n = 0, 1, 2\ldots$. (Then the hypergeometric function is just a finite degree polynomial in $\cos^2\alpha$.) Thus, we must have

$$k = \frac{3}{2} + \mu + \ell + 2n, \qquad n = 0, 1, 2\ldots, \tag{D.5}$$

and the eigenvalues of $-\Delta + \langle i\lambda \rangle + \frac{3}{4}$ are

$$E_p = -\frac{1}{4} + \left(\frac{3}{2} + \mu + p\right)^2, \qquad p = 0, 1, 2\ldots \tag{D.6}$$

Here $p = \ell + 2n$ and the degeneracy of level $E_p$ is $g(p) = \frac{(p+1)(p+2)}{2}$. Note that we are only interested in $\mu > -1$, so $E_p > 0$. The free energy

$$F_{HS^3} = \frac{1}{2}\sum_{p=0}^\infty g(p)\log\frac{E_p}{R^2}. \tag{D.7}$$

Here, we have re-instated the radius of the hemisphere $R$, since we are interested in the logarithmic term $\sim \log R$ in the free energy. We now apply the $\zeta$-function regularization to the above formal sum:

$$F_{HS^3} = -\frac{1}{2}\frac{d}{ds}\left[R^{2s}\sum_{p=0}^\infty g(p)E_p^{-s}\right] \sim -\log R\sum_{p=0}^\infty g(p)E_p^{-s}. \tag{D.8}$$

Analytic continuation to $s = 0$ is understood throughout and in the last step we've dropped a constant term. Writing $E_p = (p + \mu + 1)(p + \mu + 2)$, we use the Feynman trick:

$$
\begin{aligned}
F_{HS^3} &= -\frac{1}{2} \log R \frac{\Gamma(2s)}{\Gamma(s)^2} \sum_{p=0}^{\infty} \int_0^1 du\, u^{s-1}(1-u)^{s-1} \frac{(p+1)(p+2)}{(p+\mu+1+u)^{2s}} \\
&= -\frac{1}{2} \log R \frac{\Gamma(2s)}{\Gamma(s)^2} \int_0^1 du\, u^{s-1}(1-u)^{s-1} X(s,u,\mu),
\end{aligned}
\tag{D.9}
$$

with

$$
\begin{aligned}
X(s,u,\mu) = {}& \zeta(-2+2s, \mu+u+1) + (1-2\mu-2u)\zeta(-1+2s, \mu+u+1) \\
& - (\mu+u)(1-\mu-u)\zeta(2s, \mu+u+1).
\end{aligned}
\tag{D.10}
$$

Here $\zeta(s,a)$ is the Hurwitz $\zeta$-function (analytic continuation of $\zeta(s,a) = \sum_{p=0}^{\infty} \frac{1}{(p+a)^s}$). $X(s,u)$ is analytic near $s = 0$ in the interval $u \in [0,1]$. Therefore, the only singularities in the integral (D.9) occur at $u \to 0$, $u \to 1$. These give $1/s$ poles as $s \to 0$, which compensate the $\Gamma$-function prefactor in (D.9). Thus,

$$
F_{HS^3} = -\frac{1}{4} \log R \times (X(s=0, u=0, \mu) + X(s=0, u=1, \mu)) = \frac{1}{6}\mu^3 \log R.
\tag{D.11}
$$

# E Normal fixed point at large-$N$ in $2 < D < 4$

We begin with the non-linear $\sigma$-model

$$
S_{\inf} = \int_{x_D \geq 0} d^D x \left[ \frac{1}{2}(\partial_\mu \vec{\phi})^2 + \frac{i\lambda}{2}\left(\vec{\phi}^2 - \frac{1}{g_{\text{bulk}}}\right) \right]
\tag{E.1}
$$

at its normal boundary fixed point. We summarize the results of Ref. [25] that studied the following bulk two-point function at large-$N$:

$$
G_m(x,y) = \frac{1}{N}\left(\langle \phi^i(x)\phi^i(y)\rangle + \langle \phi^N(x)\phi^N(y)\rangle_{conn}\right).
\tag{E.2}
$$

It was shown that

$$
G_m(x,y) = \frac{\Lambda^{-\eta}}{(4x^D y^D)^{\Delta_\phi}}\left(g^0(v) + \frac{1}{N}g^1(v) + O(N^{-2})\right).
\tag{E.3}
$$

So far we have not normalized $G_m$. $\eta$ is the bulk anomalous dimension of the $\phi$ field, $\Delta_\phi = (D-2+\eta)/2$,

$$
\eta = \frac{1}{N}\frac{2(4/D-1)\Gamma(D-1)}{\Gamma(D/2-1)\Gamma(2-D/2)\Gamma(D/2)^2} + O(N^{-2}).
\tag{E.4}
$$

Here, the leading order transverse correlation function is expressed in terms of,

$$
g^0(v) = \frac{2^{D-3}\Gamma(D/2)}{\pi^{D/2}}k(v), \qquad k(v) = \frac{v^{-(D-1)}}{D-1}\,_2F_1\left(\frac{D-1}{2}, \frac{D}{2}, \frac{D+1}{2}, \frac{1}{v^2}\right).
\tag{E.5}
$$

The leading order correction in $1/N$ to $G_m$ is expressed in terms of $g^1(v)$,

$$
\begin{aligned}
g^1(v) = & -2 \int_v^\infty dv_1 (v_1^2-1)^{-D/2} \int_{v_1}^\infty dv_2 (v_2^2-1)^{D/2-1} \\
& \times \int_{v_2}^\infty dv_3 (v_3^2-1)^{-D/2} \int_{v^3}^\infty dv_4 (v_4^2-1)^{D/2-1} v(v_4) \left( g^0(v_4) + \frac{(A_\sigma^0)^2}{N} \right) \\
= & -2 \int_v^\infty dv_1 (v_1^2-1)^{D/2-1} (k(v_1) - k(v)) \\
& \times \int_{v_1}^\infty dv_2 (v_2^2-1)^{D/2-1} (k(v_2) - k(v_1)) v(v_2) \left( g^0(v_2) + \frac{(A_\sigma^0)^2}{N} \right).
\end{aligned}
\tag{E.6}
$$

Here one goes from the first to the second line using integration by parts. The function $v(v)$ is related to the $\lambda$ propagator via:

$$
D_\lambda(x,y) = \langle \lambda(x)\lambda(x') \rangle_{\text{conn}} = \frac{2}{N} \frac{1}{(x^D y^D)^2} v(v) + O(N^{-2}),
\tag{E.7}
$$

and

$$
v(v) = \frac{2^D \Gamma((D-1)/2)}{\sqrt{\pi} \Gamma(D/2-1)\Gamma(2-D/2)\Gamma(D/2-2)} \left( \frac{Q_{D-1}^{(2)}(v)}{v^2-1} + \frac{D}{D-4} \frac{Q_{D-2}^{(1)}(v)}{\sqrt{v^2-1}} \right).
\tag{E.8}
$$

The associated Legendre functions $Q$ are expressed in terms of hypergeometric functions

$$
Q_\nu^{(1)}(v) = -\frac{\Gamma(\nu+2)\sqrt{\pi}}{2^{\nu+1}\Gamma(\nu+3/2)} (v^2-1)^{1/2} v^{-\nu-2} \, {}_2F_1\left( \frac{\nu}{2}+\frac{3}{2}, \frac{\nu}{2}+1, \nu+\frac{3}{2}, v^{-2} \right),
\tag{E.9}
$$

$$
Q_\nu^{(2)}(v) = \frac{\Gamma(\nu+3)\sqrt{\pi}}{2^{\nu+1}\Gamma(\nu+3/2)} (v^2-1) v^{-\nu-3} \, {}_2F_1\left( \frac{\nu}{2}+2, \frac{\nu}{2}+\frac{3}{2}, \nu+\frac{3}{2}, v^{-2} \right).
\tag{E.10}
$$

The constant $A_\sigma^0$ is related to the one-point function of $\phi_N$ (not normalized) via:

$$
\langle \phi_N(x) \rangle = \frac{A_\sigma^0}{(2x^D)^{(D-2)/2}} + O(N^{-1/2}),
\tag{E.11}
$$

and is given by

$$
\frac{(A_\sigma^0)^2}{N} = -\frac{\Gamma(D-1)\Gamma(1-D/2)}{4\pi^{D/2}}.
\tag{E.12}
$$

Our goal is to extract $a_\sigma^2$, $b_t^2$ and, thereby, $s^2$ from $G_m(x,y)$. Based on the bulk OPE,

$$
\phi_{\text{norm}}^a(x) \phi_{\text{norm}}^a(y) = \frac{N}{(x-y)^{2\Delta_\phi}} \left( 1 + \lambda_{\phi\phi\epsilon}(x-y)^{\Delta_\epsilon} \epsilon(y) + \dots \right),
\tag{E.13}
$$

we have

$$
\begin{aligned}
G_m(x,y) = & \frac{C\Lambda^{-\eta}}{(4x_D y_D)^{\Delta_\phi}} \frac{1}{((v-1)/2)^{\Delta_\phi}} \\
& \times \left[ 1 - \left( \frac{v-1}{2} \right)^{\Delta_\phi} \frac{a_\sigma^2}{N} + \lambda_{\phi\phi\epsilon} a_\epsilon \left( \frac{v-1}{2} \right)^{\Delta_\epsilon/2} + \dots \right], \qquad v \to 1,
\end{aligned}
\tag{E.14}
$$

where $\langle \epsilon(x) \rangle = \frac{a_\epsilon}{(2x^D)^{\Delta_\epsilon}}$. The unnormalized field $\phi$ appearing in (E.2) is related to the normalized field $\phi_{\text{norm}}$ via $\phi = \sqrt{C}\Lambda^{-\eta/2}\phi_{\text{norm}}$. Thus, the normalization constant $C$ and $a_\sigma^2$ can be

read-off from the $\nu \to 1$ limit of $G_m$. We write,

$$C = C_0\left(1 + \frac{c_1}{N} + O\left(N^{-2}\right)\right), \qquad C_0 = \frac{\Gamma(D/2-1)}{4\pi^{D/2}},$$
$$a_\sigma^2 = (a_\sigma^0)^2\left(1 + \frac{r}{N} + O\left(N^{-2}\right)\right), \qquad (a_\sigma^0)^2 = -\frac{N\Gamma(D-1)\Gamma(1-D/2)}{\Gamma(D/2-1)}. \qquad \text{(E.15)}$$

with $c_1$ and $r$ - to be determined.

Based on the boundary OPE, (115), (116),

$$G_m(x,y) = \frac{C\Lambda^{-\eta}}{(4x_D y_D)^{\Delta_\phi}}\left(1 - \frac{1}{N}\right)\frac{2^{D-1}b_t^2}{\nu^{D-1}}, \qquad \nu \to \infty. \qquad \text{(E.16)}$$

We note that $g^1(\nu)$ decays as $1/\nu^D$ for $\nu \to \infty$, thus,

$$b_t^2 = \frac{D/2-1}{D-1}\left(1 + \frac{1}{N}(1-c_1) + O\left(N^{-2}\right)\right). \qquad \text{(E.17)}$$

We perform the integral in (E.6) numerically for $\nu \to 1$ in order to extract $c_1$ and $r$. We begin by discussing the behavior of $g^1(\nu)$ for $\nu \to 1$. We have

$$g^0(\nu) + \frac{(A_\sigma^0)^2}{N} = 2^{D-2}C_0(\nu^2-1)^{1-D/2}\nu^{-1}\,_2F_1\left(1,1/2,2-D/2,1-\nu^{-2}\right). \qquad \text{(E.18)}$$

Then

$$(\nu^2-1)^{D/2-1}\nu(\nu)\left(g^0(\nu) + \frac{(A_\sigma^0)^2}{N}\right) = \frac{q_{-2}}{(\nu-1)^2} + \frac{q_{-1}}{\nu-1} + \dots, \qquad \nu \to 1, \qquad \text{(E.19)}$$

with

$$q_{-2} = \frac{2^{2D-5}\Gamma((D-1)/2)}{\pi^{(D+1)/2}\Gamma(2-D/2)\Gamma(D/2-2)}, \qquad q_{-1} = -\frac{(D-1)(D-2)^2}{2(D-4)}q_{-2}. \qquad \text{(E.20)}$$

The $\nu_4$ integral in the second line of (E.6) then gives

$$\int_{\nu_3}^{\infty}d\nu_4(\nu_4^2-1)^{D/2-1}\nu(\nu_4)\left(g^0(\nu_4) + \frac{(A_\sigma^0)^2}{N}\right) = \frac{q_{-2}}{\nu_3-1} - q_{-1}\log(\nu_3-1) + p + \dots, \quad \nu_3 \to 1,$$
$$\text{(E.21)}$$

where $p$ is a constant that we can only determine numerically. Performing the integrals over $\nu_3$ and $\nu_2$ in (E.6)

$$\int_{\nu_1}^{\infty}d\nu_2(\nu_2^2-1)^{D/2-1}\int_{\nu_2}^{\infty}d\nu_3(\nu_3^2-1)^{-D/2}\int_{\nu^3}^{\infty}d\nu_4(\nu_4^2-1)^{D/2-1}\nu(\nu_4)\left(g^0(\nu_4) + \frac{(A_\sigma^0)^2}{N}\right)$$
$$= \int_{\nu_1}^{\infty}d\nu_2(\nu_2^2-1)^{D/2-1}\int_{\nu_2}^{\infty}d\nu_3(\nu_3^2-1)^{D/2-1}(k(\nu_3)-k(\nu_2))\nu(\nu_3)\left(g^0(\nu_3) + \frac{(A_\sigma^0)^2}{N}\right)$$
$$= \frac{1}{2}\left[-\frac{2q_{-2}}{D}\log(\nu_1-1) + p' - \left(\frac{1}{D/2-1}\left(p - \frac{q_{-2}D}{4} - \frac{q_{-1}(D-4)}{D-2}\right)\right.\right.$$
$$\left.\left.+ \frac{q_{-2}(D/2-1)}{D}\right)(\nu_1-1) + \frac{q_{-1}}{D/2-1}(\nu_1-1)\log(\nu_1-1)\right] + \dots, \qquad \nu_1 \to 1. \quad \text{(E.22)}$$

$p'$ is a constant that we can only determine numerically. Finally,

$$g^1(v) = -2^{-D/2}(v-1)^{1-D/2}\left\{-\frac{2q_{-2}}{D(D/2-1)}\log(v-1) + \frac{1}{D/2-1}\left(p' - \frac{2q_{-2}}{D(D/2-1)}\right)\right.$$
$$+ \frac{(v-1)\log(v-1)}{D/2-2}\left(\frac{q_{-1}}{D/2-1} + \frac{q_{-2}}{2}\right)$$
$$+ \frac{(v-1)}{D/2-2}\left[-\frac{1}{D/2-1}\left(p - \frac{q_{-2}D}{4} - \frac{q_{-1}(D-4)}{D-2}\right) + \frac{1}{D/2-2}\left(\frac{q_{-1}}{D/2-1} + \frac{q_{-2}}{2}\right)\right.$$
$$\left.\left.-\frac{q_{-2}(D/2-1)}{D} - \frac{p'D}{4}\right]\right\} + p''. \tag{E.23}$$

Again, $p''$ is a constant to be determined numerically. Matching to (E.14),

$$c_1 = -\frac{p'}{(D-2)C_0} - \frac{\eta N}{2}\left(\frac{1}{D/2-1} + \log 2\right), \tag{E.24}$$

$$r = -\frac{p''}{(A_\sigma^0)^2/N} - c_1, \tag{E.25}$$

where $r$ determines the $1/N$ correction to $a_\sigma^2$, Eq. (E.15), while $c_1$ determines the $1/N$ correction to $b_t^2$, Eq. (E.17). Finally, for the constant $s^2$, (114),

$$s^2 = s_0^2\left(1 + \frac{f}{N}\right), \qquad f = -\frac{p''}{(A_\sigma^0)^2/N} - 1, \qquad s_0^2 = -\frac{N\Gamma(D)}{4\pi^{D-1}\sin(\pi D/2)}, \tag{E.26}$$

where $f$ is the function we introduced in (143). We want to determine $f'(D=3)$.

We proceed by first evaluating the integral (E.21) numerically for $v \to 1$ to determine $p$. We then evaluate the integral in the second line of (E.22) to determine $p'$. Finally, we evaluate the integral (E.6) to determine $p''$. The resulting values of the coefficients of $1/N$ corrections to $a_\sigma^2$, Eq. (E.15), $b_t^2$, Eq. (E.17), and $s^2$, Eq. (E.26), are shown in Fig. 14. The numerical results are in good agreement with the analytical result at $D=3$: $r(D=3) = 1 - c_1(D=3) = 1 - \eta(D=3)/2$, so that $f(D=3) = 0$. [1] They are also in good agreement with the results of $2+\epsilon$ and $4-\epsilon$ expansions [1, 49]:

$$a_\sigma^2 = N\left(1 + \frac{\pi^2}{12}\frac{N-1}{N-2}\epsilon^2 + O(\epsilon^3)\right), \tag{E.27}$$

$$b_t^2 = \frac{\epsilon N}{2(N-2)}\left(1 - \epsilon\frac{N-1}{N-2} + O(\epsilon^2)\right), \qquad D = 2+\epsilon,$$

$$a_\sigma^2 = \frac{4(N+8)}{\epsilon}\left(1 - \frac{N^2+31N+154}{(N+8)^2}\epsilon + O(\epsilon^2)\right), \tag{E.28}$$

$$b_t^2 = \frac{1}{3}\left(1 - \epsilon\frac{N+9}{6(N+8)} + O(\epsilon^2)\right), \qquad D = 4-\epsilon. \tag{E.29}$$

To determine $f'(D=3)$ we fit $f(D)$ in the window $2.95 < D < 3.05$ to a quadratic function. We obtain $f'(D=3) = 3.67(1)$. Here, the error bar is conservatively estimated by increasing the range of the fit to $2.9 < D < 3.1$.

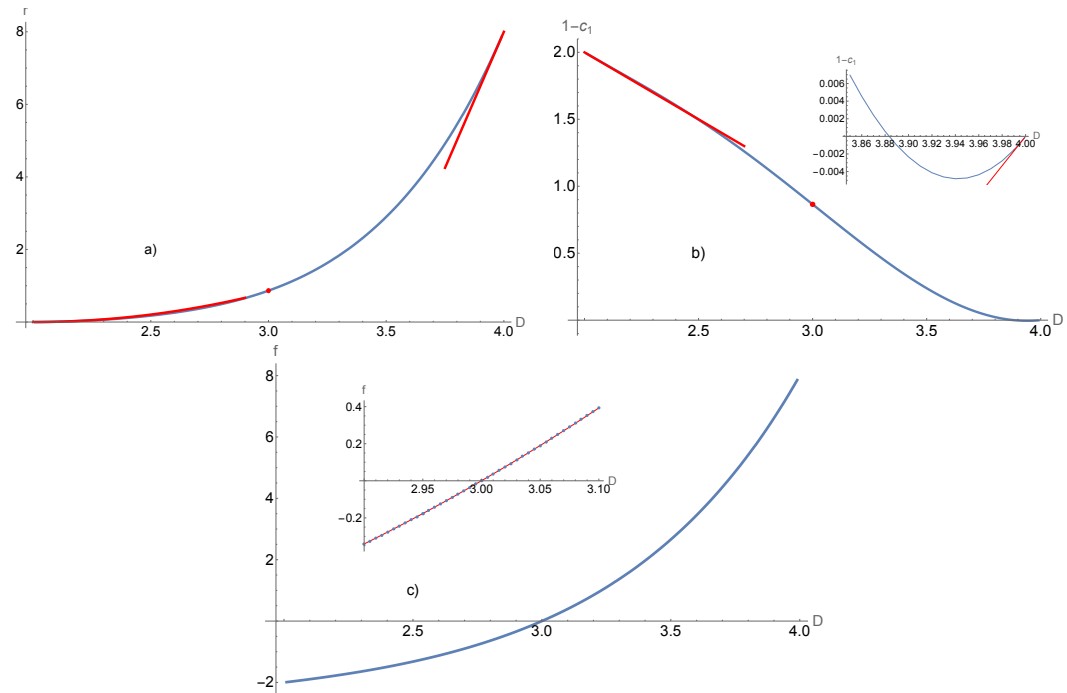

Figure 14: a) The coefficient $r$ of the $1/N$ correction to $a_\sigma^2$, Eq. (E.15).
b) The coefficient $1 - c_1$ of the $1/N$ correction to $b_t^2$, Eq. (E.17). For both of a) and
b) the red lines are asymptotes expected from $2 + \epsilon$ and $4 - \epsilon$ expansions; the solid
dot at $D = 3$ is the analytical calculation.
Bottom: c) The coefficient $f$ of the $1/N$ correction to $s^2$, Eq. (E.26). Inset: a quadratic
fit to $f(D)$ for $D$ near 3.

# F  Useful Integrals

The following integrals are used in this work:

$$I_+(q, \theta) = \int_\theta^\infty \frac{e^{q\alpha}\mathrm{d}\alpha}{(\cosh\alpha + 1)^2} = -\frac{2}{3}\frac{e^{(1+q)\theta}(q - 1 + e^\theta(1+q))}{(1 + e^\theta)^3} + \frac{2}{3}\pi q(1 - q^2)\csc(\pi q) \quad (\text{F.1})$$
$$+ \frac{2}{3}(q-1)e^{(1+q)\theta}{}_2F_1\left(2, 1 + q, 2 + q, -e^\theta\right),$$

$$I_-(q, \theta) = \int_\theta^\infty \frac{e^{q\alpha}\mathrm{d}\alpha}{(\cosh\alpha + 1)^2} = \frac{2}{3}\Bigg[q(q^2 - 1)\beta(e^{-\theta}, -1 - q, 0)+ \quad (\text{F.2})$$
$$\frac{e^{(3+q)\theta}((-2q + 1)(q - 1) + (3 - 3q + 2q^2)\cosh\theta + (q - 3)\sinh\theta)}{(e^\theta - 1)^3}\Bigg],$$

$$\tilde{I}_+(q, \theta) = \int_\theta^\infty \frac{\alpha e^{q\alpha}\mathrm{d}\alpha}{(\cosh\alpha + 1)^2} \quad (\text{F.3})$$
$$= \frac{2}{3}\pi\csc(\pi q)(1 - 3q^2 + \pi q(q^2 - 1)\cot(\pi q))-$$
$$\frac{2}{3}\frac{2\theta - (1 + e^\theta)(1 + \theta(3 + q)) + (1 + e^\theta)^2(1 + q(2 + \theta + \theta q))}{e^{-\theta q}(e^\theta + 1)^3}-$$
$$\frac{2e^{\theta q}}{3}\left[q(q^2 - 1)\Phi(-e^\theta, 2, q) + (1 + q(\theta - 3q - \theta q^2))\Gamma(q){}_2\tilde{F}_1(1, q, 1 + q, -e^\theta)\right],$$

$$\tilde{I}_-(q,\theta) = \int_\theta^\infty \frac{\alpha e^{q\alpha} d\alpha}{(\cosh\alpha - 1)^2} \tag{F.4}$$

$$= \frac{2}{3}(-1)^{-q}\pi(1 - 3q^2 + i\pi q(q^2 - 1) + \pi q(q^2 - 1)\cot(\pi q))\csc(\pi q) -$$

$$\frac{1}{3}e^{q\theta}\left(\frac{1}{q} - q - \theta + q^2\theta + \frac{\sinh\theta(\theta - q(2 + q\theta)) - \theta(q + \coth(\theta/2)) - 1}{\cosh\theta - 1}\right) -$$

$$\frac{1}{3}(1 - 3q^2)\beta(e^\theta, 1 + q, 0) - \frac{1}{3}(1 + q(2\theta - q(3 + 2q\theta)))\beta(e^\theta, q, 0) -$$

$$\frac{2}{3}q(q^2 - 1)\Phi(e^\theta, 2, q)e^{q\theta}.$$

The latter two integrals were evaluated using results from App. G.

We now compute the behavior of these integrals as $\theta \to 0$:

$$I_+(q,\theta) \approx \frac{1}{6}\left\{2 + q(3 + 2q) + 2q(q^2 - 1)[H(1/2 - q/2) - H(-q/2)]\right\} - \frac{\theta}{4}, \tag{F.5}$$

$$I_-(q,\theta) \approx \frac{4}{3\theta^3} + \frac{2q}{\theta^2} + \frac{2q^2}{\theta} - \frac{2}{3\theta} - \frac{2}{3}q(q^2 - 1)\log\theta + \tag{F.6}$$

$$\frac{1}{18}(-6 + q(-25 + 2q(9 + 11q)) - 12(q^3 - q)H(-q - 2)) -$$

$$\frac{1}{180}(11 + 30q^2(q^2 - 2))\theta, \tag{F.7}$$

$$\tilde{I}_+(q,\theta) \approx \frac{1}{6}\Big[3 + 4q + (2 - 6q^2)\psi(1 - q/2) + (-2 + 6q^2)\psi(3/2 - q/2) + \tag{F.8}$$

$$q(-1 + q^2)(\psi^{(1)}(1 - q/2) - \psi^{(1)}(3/2 - q/2))\Big],$$

$$\tilde{I}_-(q,\theta) \approx \frac{2}{\theta^2} + \frac{4q}{\theta} - (2/3)\theta q(-1 + q^2) - \tag{F.9}$$

$$\frac{1}{18}\Big(13 - 12\gamma_E + 12q + 6(-11 + 6\gamma_E)q^2 -$$

$$12\pi^2 q(-1 + q^2)\csc^2(\pi q) + 12\pi(3q^2 - 1)\cot(\pi q) + \tag{F.10}$$

$$12(-1 + 3q^2)\log\theta + 12(-1 + 3q^2)\psi(q) + 12q(-1 + q^2)\psi^{(1)}(q)\Big).$$

# G  Asymptotic Behavior of Hurwitz Lerch Transcendent

We are interested in the asymptotic behavior of

$$\Phi(z, 2, q),$$

where $z \to \pm\infty$. There are four cases to consider per our calculations.

## G.1  Case 1

Consider $0 < q < 2$ and $z > 0$. Per Ref. [50],

$$\Phi(e^\alpha, 2, q) = \frac{1}{\Gamma(s)}\left[\sum_{n=0}^\infty \frac{A_n(e^\alpha, 2, q)}{e^{(n+1)\alpha}} + e^{-q(\alpha+i\pi)}(B_0(2, q)(\alpha + i\pi) + B_1(2, q))\right], \tag{G.1}$$

where

$$A_n(z, 2, q) = \frac{\Gamma(2, (q - n - 1)\log(-z)) - 1}{(q - n - 1)^2}, \tag{G.2}$$

$$B_0(2, q) = \frac{\psi(q/2 + 1/2) - \psi(q/2)}{2}, \quad B_1(2, q) = \frac{\psi^{(1)}(2, q/2) - \psi^{(1)}(2, q/2 + 1/2)}{4}. \tag{G.3}$$

Note that

$$\frac{A_n(e^\alpha, 2, q)}{e^{(n+1)\alpha}} \approx \frac{-e^{-(n+1)\alpha} - (-1)^n e^{-q(\alpha + i\pi)}(1 - (\alpha + i\pi)(1 + n - q))}{(q - n - 1)^2}. \tag{G.4}$$

Using that $0 < q < 2$, the sum is asymptotically

$$\sum_{n=0}^{\infty} \frac{A_n(e^\alpha, 2, q)}{e^{(n+1)\alpha}} \approx \frac{-e^{-\alpha}}{(q-1)^2} - \frac{e^{-2\alpha}}{q^2} + e^{-q(\alpha + i\pi)}[B_1(2, 1/2 - q/2) - (\alpha + i\pi)B_0(2, 1/2 - q/2)]. \tag{G.5}$$

Then,

$$\Phi(e^\alpha, 2, q) \approx -\frac{e^{-\alpha}}{(q-1)^2} - \frac{e^{-2\alpha}}{q^2} + e^{-q(\alpha + i\pi)}\pi(\alpha + i\pi + \pi\cot(\pi q))\csc(\pi q). \tag{G.6}$$

## G.2 Case 2

Now consider $-2 < q < 0$ and $z > 0$. We know that $2 + q > 0$, so we can apply Eq. (G) to $\Phi(e^\alpha, 2, 2 + q)$. Then, from the series definition of $\Phi$,

$$\Phi(e^\alpha, 2, q) = \frac{1}{q^2} + \frac{e^\alpha}{(1+q)^2} + e^{2\alpha}\Phi(e^\alpha, 2, 2+q) \approx e^{-q(\alpha + i\pi)}\pi(\alpha + i\pi + \pi\cot(\pi q))\csc(\pi q). \tag{G.7}$$

## G.3 Case 3

Consider $0 < q < 2$ and $z < 0$. We again use Ref. [50]:

$$\Phi(-e^\alpha, 2, q) = \frac{1}{\Gamma(s)}\left[(-1)^{n+1}\sum_{n=0}^{\infty} \frac{A_n(-e^\alpha, 2, q)}{e^{(n+1)\alpha}} + e^{-q\alpha}(B_0(2, q)\alpha + B_1(2, q))\right], \tag{G.8}$$

where

$$A_n(z, 2, -e^\alpha) = \frac{\Gamma(2, (q - n - 1)\alpha) - 1}{(q - n - 1)^2}, \tag{G.9}$$

$$B_0(2, q) = \frac{\psi(q/2 + 1/2) - \psi(q/2)}{2}, \quad B_1(2, q) = \frac{\psi^{(1)}(2, q/2) - \psi^{(1)}(2, q/2 + 1/2)}{4}. \tag{G.10}$$

Note that

$$\frac{(-1)^{n+1}A_n(-e^\alpha, 2, q)}{e^{-(n+1)\alpha}} \approx \frac{(-1)^n e^{-(n+1)\alpha} + (-1)^{n+1}e^{-\alpha q}(1 - \alpha(1 + n - q))}{(q - n - 1)^2}. \tag{G.11}$$

Using that $0 < q < 2$, the sum is asymptotically

$$\sum_{n=0}^{\infty} \frac{(-1)^{n+1}A_n(-e^\alpha, 2, q)}{e^{(n+1)\alpha}} \approx \frac{e^{-\alpha}}{(q-1)^2} - \frac{e^{-2\alpha}}{q^2} + e^{-q\alpha}[\alpha B_0(2, 1/2 - q/2) - B_1(2, 1/2 - q/2)]. \tag{G.12}$$

Then,

$$\Phi(-e^\alpha, 2, q) \approx \frac{e^{-\alpha}}{(q-1)^2} - \frac{e^{-2\alpha}}{q^2} + e^{-q\alpha}\pi(\alpha + \pi\cot(\pi q))\csc(\pi q). \tag{G.13}$$

### G.4 Case 4

Finally, consider $-2 < q < 0$ and $z < 0$. Using the series expansion,

$$\Phi(-e^{\alpha}, 2, q) \approx e^{-q\alpha} \pi(\alpha + \pi \cot(\pi q)) \csc(\pi q). \tag{G.14}$$

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
