# Peer review of "A plane defect in the 3d O(N) model"

_SciPost Physics, doi:SciPost Phys. 15, 090 (2023)_

## Round 1 · Referee Report · Antonio Antunes (Referee 1) · 2023-5-12

Strengths

1 - The paper extends a recent important result on the boundary criticality of the O(N) model to the case of a critical interface. The result is qualitatively different from the boundary case, providing evidence for the existence of an extraordinary-log class for all N>=2, which is remarkable.

2 - It also includes relevant results on how the possible interface RG flows are consistent with monotonicity theorem, as well as providing strong evidence for the form of the phase diagram in the boundary case, suggested in previous work.

3 - A very well-written introduction gives a good summary of all the different results and provides a coherent picture of the full paper.

Weaknesses

1 - Each section feels a bit too independent of the others, since they cover somewhat different aspects. This is mitigated by the excellent introduction.

Report

The paper presents new and interesting results on both conformal boundaries and conformal interfaces of the critical 3d O(N) model. It gives evidence for the existence of an extraordinary-log interface universality class for all N>=2, in opposition to the boundary case where this only happens in a window 2<=N<N_c .

The manuscript complements this result with an in-depth analysis of interface RG flows using monotonicity theorems, as well as computing a key quantity (at large N) in the boundary case, which gives evidence for the detailed form of the phase diagram in the 2<=N<N_c range.

The paper is overall of excellent quality and I recommend it for Publication.

I do have some questions and suggestions that might help further clarify certain points, but my recommendation for publication is not contingent on the authors addressing them.

Requested changes

1 - In page 4 below eq 1.3, it is said that "believed that the scaling dimension ∆ϵ < 2 for all finite N ≥ 1". It might be worth pointing out the existence of rigorous bootstrap bounds which show this for many values of N, including moderately large N.

2 - In page 16 it is said that "the
interface ordinary fixed point is equivalent to two decoupled boundary ordinary fixed points for each
side of the interface". Reminding the reader of eq. 1.4 could be helpful.

3 - It is used in multiple occasions that the special interface transition is essentially equivalent to no defect for all N. On the other hand, the boundary special transition has non-trivial anomalous dimensions with the respect to the bulk operators. Is there a simple way to argue for the triviality of the interface with an action similar to 1.4 or 2.2 for the special boundary class?

4 - The folding construction is used a few times in the text. Could one have anticipated eq. 5.37 from 3.34 + the folding argument?

5 - Reference https://inspirehep.net/literature/1837673 computes conformal data for the O(N) conformal interface in the epsilon expansion (for the ordinary and special transitions ). Maybe these results could be mentioned. Perhaps some of the them are useful to compare with?

---

## Round 1 · Referee Report · Anonymous (Referee 2) · 2023-6-8

Strengths

  • Extremely interesting topic. The paper focuses on the phase diagram of defect CFTs in the O(N) model, and in particular, on an unusual universality class that exhibits logarithmic behavior in the spin two-point function.

  • Detailed RG and perturbative calculations that clarify the picture, and serve as check for several of the claims made in the paper.

  • Nice pictures and plots.

Weaknesses

None

Report

This paper studies the ''extraordinary log'' universality class for defects in the O(N) model. This class is characterized by a spin two-point function that falls off logarithmically, as opposed to the more standard power-law behavior. This extremely interesting universality class had been studied for BCFT in the O(N) model, and the analysis is now extended to the case of an interface. The main result of the paper is that log universality class is expected for any N>=2, while for BCFT it was constrained to a range.

The paper contains careful perturbative calculations with a lot of details. In addition to studying the phase diagram, the authors also check the consistency of their results with the defect version of the a-theorem. As far as I can tell, everything looks solid.

This is an excellent paper, full of interesting results. I strongly recommend it for publication

Requested changes

No changes from my part

---

## Round 2 · Referee Report · Antonio Antunes (Referee 1) · 2023-7-10

Report

I am happy with the changes and recommend the paper for publication.

---

## Round 2 · List of Changes

We would like to thank the Referee for the careful reading of our manuscript and for their comments.

1 - In page 4 below eq 1.3, it is said that "believed that the scaling dimension ∆ϵ < 2 for all finite N ≥ 1". It might be worth pointing out the existence of rigorous bootstrap bounds which show this for many values of N, including moderately large N.

Upon the Referee’s suggestion, below Eq. (1.3) we have added a citation to Ref [12] (a bootstrap study identifying the O(N) islands) and Ref [13] (a review of the large-N results).

2 - In page 16 it is said that "the interface ordinary fixed point is equivalent to two decoupled boundary ordinary fixed points for each side of the interface". Reminding the reader of eq. 1.4 could be helpful.

We have added a reference to Eq. (1.4), as suggested by the Referee.

3 - It is used in multiple occasions that the special interface transition is essentially equivalent to no defect for all N. On the other hand, the boundary special transition has non-trivial anomalous dimensions with the respect to the bulk operators. Is there a simple way to argue for the triviality of the interface with an action similar to 1.4 or 2.2 for the special boundary class?

We can start with two decoupled boundaries each at its special fixed point (assuming N < N_c ~ 5). However, the dimension of the boundary O(N) vector at the boundary special fixed point is quite small (please see table I in Ref. [1]). Even for N=1, this dimension is ~0.36 and the dimension decreases with N. Thus, the analog of the coupling u in Eq. (1.4) is highly relevant. We cannot track the RG flow induced by u, however, the natural guess is that it is towards the trivial interface, which, importantly, has only a single relevant O(N) singlet perturbation.

4 - The folding construction is used a few times in the text. Could one have anticipated eq. 5.37 from 3.34 + the folding argument?

We were very intrigued by the apparent doubling of the shift of the g^3 coefficient in the beta-function from the single boundary to the interface. One way to uncover the origin of this doubling would be to pursue the strategy explained on page 24:

``From the form of the action (5.1) a direct computation of the coefficient b in beta(g) requires the knowledge of the four-point function of the tilt operator t_i at the normal fixed point. (This should be compared to the computation of the coefficient alpha_{bound}, which relies only on the two-point function of t_i and the knowledge of the coefficient s. In addition, a number of higher order counter-terms in the action, omitted in
Eq. (5.1), such as e.g. delta L_{bound} ~ pi^2 pi_i t_i, would have to be fixed by the requirement of O(N) invariance. We do not pursue this route to computing b here.”

Instead of pursuing this plan, we chose to extract the g^3 coefficient in the beta-function from the boundary order parameter dimension at the special transition in d=3+epsilon. However, it would be interesting to pursue the calculation sketched above in future work.

5 - Reference https://inspirehep.net/literature/1837673 computes conformal data for the O(N) conformal interface in the epsilon expansion (for the ordinary and special transitions ). Maybe these results could be mentioned. Perhaps some of the them are useful to compare with?

We thank the Referee for pointing out this paper. Since we do not use the OPE coefficients at the ordinary/special transition anywhere in our paper, we don’t see a natural place to cite this reference (we also don’t cite older papers on the epsilon expansion for the ordinary/special transition, since we don’t use the corresponding results.)

---

## Editorial Decision

published